# Plio–Quaternary Structural Evolution of the Outer Sector of the Marche Apennines South of the Conero Promontory, Italy

**Mario Costa [1], Jessica Chicco [2], Chiara Invernizzi [3],\*, Simone Teloni [3] and Pietro Paolo Pierantoni [3]**

[1] Via Selvelli, 6, 61032 Fano, Italy; ioset@libero.it
[2] Department of Earth Sciences, University of Turin, via Valperga Caluso, 35, 10125 Torino, Italy; jessica.chicco@unito.it
[3] School of Science and Technology, Geology Division, University of Camerino, via Gentile III da Varano, 62032 Camerino, Italy; simone.teloni@unicam.it (S.T.); pietropaolo.pierantoni@unicam.it (P.P.P.)
\* Correspondence: chiara.invernizzi@unicam.it; Tel.: +39-3288604234

**Abstract:** Some new results and preliminary remarks about the Plio–Quaternary structural and evolutionary characteristics of the outer Marche Apennines south in the Conero promontory are presented in this study. The present analysis is based on several subsurface seismic reflection profiles and well data, kindly provided by ENI S.p.A. and available on the VIDEPI list, together with surface geologic–stratigraphic knowledge of Plio–Quaternary evolution from the literature. Examples of negative vs. positive reactivation of inherited structures in fold and thrust belts are highlighted. Here, we present an example from the external domain of the Marche Apennines, which displays interesting reactivation examples from the subsurface geology explored. The study area shows significant evolutionary differences with respect to the northern sector of the Marche region previously investigated by the same research group. The areal distribution of the main structures changes north and south of the ENE–WSW oriented discontinuity close to the Conero promontory. Based on the old tripartite classification of the Pliocene, the results of this work suggest a strong differential subsidence with extension occurring during the Early Pliocene and principal compressive deformation starting from the Middle Pliocene and decreasing or ceasing during the Quaternary. The main structure in this area is the NNW–SSE Coastal Structure, which is composed of E-vergent shallow thrusts and high-angle deep-seated normal faults underneath. An important right-lateral strike–slip component along this feature is also suggested, which is compatible with the principal NNE–SSW shortening direction. As mentioned, the area is largely characterized by tectonic inversion. Starting from Middle Pliocene, most of the Early Pliocene normal faults became E-vergent thrusts.

**Keywords:** Plio–Quaternary evolution; outer Marche Apennines; seismic reflection profiles; tectonic inversion; Coastal Structure; extensional and contractional deformation

## 1. Introduction

In the Apennines of Italy, and especially the Adriatic foreland domain, it is possible to infer the foreland deformation process and explore the impacts of inherited faults and basins on the subsequent evolution thanks to the milder deformation in the area and the good geological and geophysical record documenting an interaction between normal, thrust, and strike–slip faults.

Foreland domains are often affected by inherited rift-related or flexure-related synsedimentary normal faults becoming involved in the advancing fold-and-thrust belt. This introduces an element of further complications into the evolution of the foredeep systems subsequently involved in the mountain belts, as evidenced by numerous studies in different contexts, such as the Northern Apennines, Po Plain, and South-Eastern Pyrenean foreland basins ([1–4], among others).

The tectonic and structural features of the Umbria-Marche Apennines (Figure 1) are widely described in the literature, and several models have been proposed. The most

important model is found in [5], which proposes a thin-skinned imbricate belt detached above the crystalline basement (see also [6]). This model indicates strong shortening (in the order of hundreds of kilometres) and important repetitions of the sedimentary cover. Further studies on the geometries and evolution of the outer Marche sector, as well as their extent, style, and age of deformation, are thoroughly reported in many works. Among others, [7–13] mainly focus on stratigraphic record, geological setting, and sedimentary evolution; [14] on the anatomy of the Apennine orogen; [15–20] on the structural and deformation style; and [14,21] on the role of inherited structures and tectonic inversion.

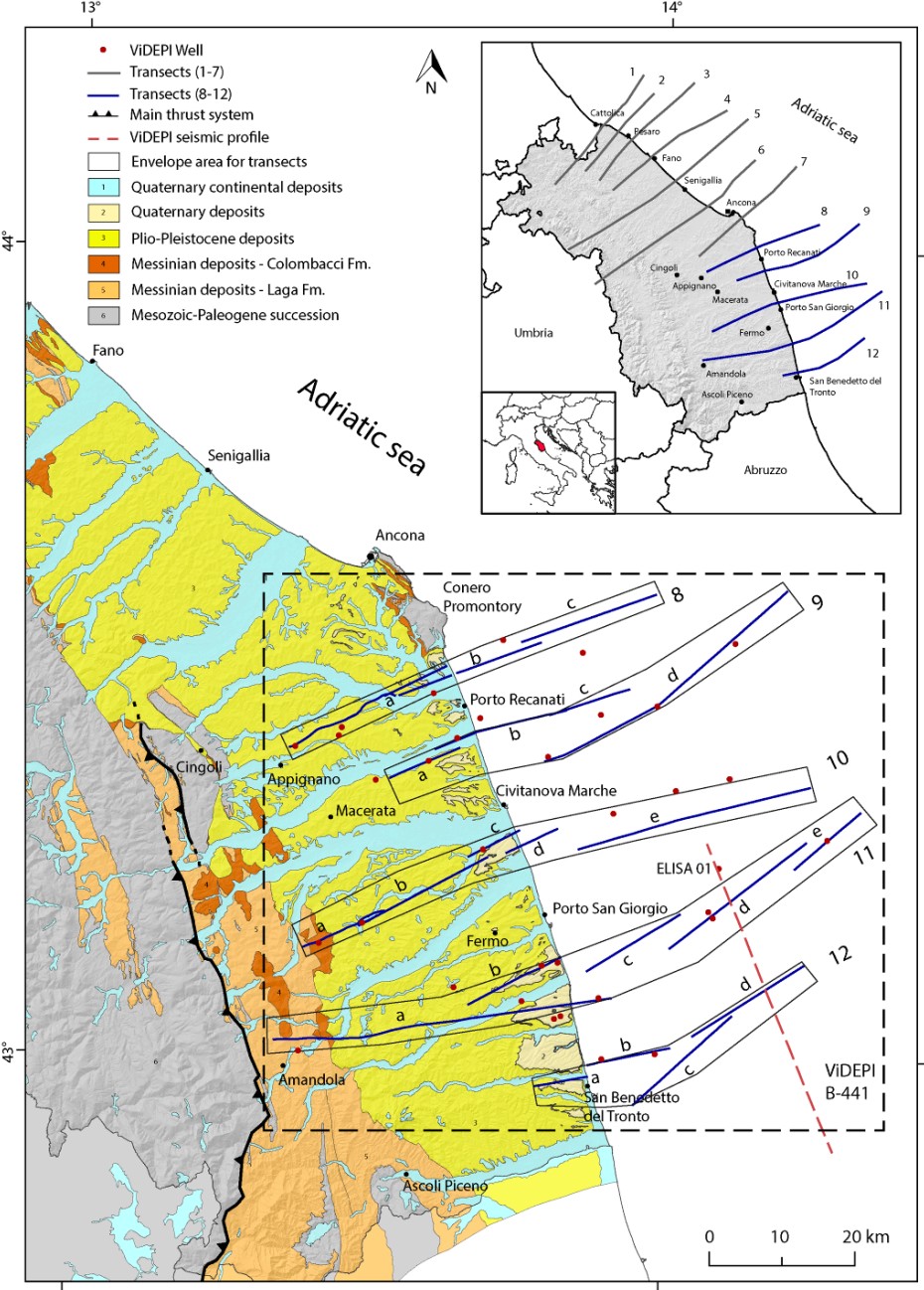

**Figure 1.** On shore schematic geological map of the Marche region (modified from [22]). The work in [23] was considered for the thrust location. Dashed square: study area; dashed red line: ViDEPI seismic profile B-441 with Elisa 1 well (Figure 3). Inset: geographic location of the study and location of the Transects. Transects: results for 1 to 7 are published in [24,25]. Transects 8 to 12: this work.

The acquisition of new data (such as the CROP project, well stratigraphy and seismic reflection profiles, and sedimentological and paleo-thermometric data) have shed new light on the evolution of the area and introduced models that indicate the crystalline basement's involvement (thick-skinned model) and the reactivation of inherited faults (inversion tectonic model). As a main outcome, the amount of shortening affecting this area was progressively reduced from hundreds of kilometres to tens of kilometres [26].

New observations about onshore and offshore outcropping and buried Neogene–Quaternary structures, as well as their possible implications for deep geothermal fluid circulation, were recently integrated into the tectonic framework of the northern outer Marche Apennines [24,25]. These studies highlighted new findings mainly characterized by the presence of positive flower structures to be considered as common features along the whole outer sector of the Northern Apennine chain [24]. This suggests the more relevant influence of strike–slip kinematics in recent times, with implications for seismic assessment and deep fluid circulation [25].

The southern sector of the outer Marche Apennines has been long investigated by authors who addressed specific features in this area as related to a complex foreland–foredeep geometry. In particular, several works explore the influence of thrust-system propagation on the distribution of sedimentary sequences, their 3D geometric organization, and the burial and exhumation history of these units [27–31]. These features were identified as the link between the inner, uplifted, and Early Miocene Apennine fold-and-thrust belt and the outer and younger belt to the east [31]. The interpretations of integrated structural and stratigraphic studies indicate this to be the result of turbidite deposition in a complex foredeep, strongly affected by tectonic activity and Messinian–Pliocene climate changes ([29,32] and references therein).

This paper represents a continuation of the above-mentioned studies [24,25] and aims at highlighting the significant structural and depositional differences between the northern and southern outer Marche Apennine, as well as discussing the timing and style of deformation in the outermost sector of the belt toward the Adriatic foreland, where milder deformation and mainly buried structures are present.

To this end, a detailed study along the sector south of the Conero promontory to S. Benedetto del Tronto was conducted (Figure 1) using seismic reflection profile interpretations and well data for hydrocarbon purposes, kindly provided by ENI S.p.A. Available online data: https://www.videpi.com/videpi/sismica/sismica.asp (accessed on 30 March 2021), and published studies further contributed to acquiring complete information and enriching the results in our previous works.

## 2. Geological Setting

At the continental scale, the Alps and Apennines orogens are located in the hanging wall of two opposite subduction zones. The Alps resulted from the Cretaceous to present via the European plate being subducted beneath the Adriatic plate to the east, whereas the Apennines resulted from the Eocene to the present via subduction of the Adriatic plate to the west ([33] and references therein). The Adriatic plate itself is also subducted below the Dinarides in its easternmost part [33,34].

The arcuate-shaped, NE-verging Umbria–Marche Apennines form the external part of the Northern Apennines foreland fold-and-thrust belt (see [35] and references therein). This belt resulted from the convergence between a mosaic of minor blocks of the Africa–Eurasia plates, such as the European Corsica–Sardinia plate to the West and the African–Adria plate to the east ([36–39] and references therein).

In the Umbria–Marche area, starting from the Miocene, the previously rifted and telescoped African-bearing continental margin was involved in the compressive phase. Here, different styles and degrees of the positive inversion of pre-orogenic faults controlled the location, geometry, and evolution of compressive structures in several cases [16,28,40–44]. In addition, the inner portion of the chain was involved in the Late Miocene to present day extension [14,23,45,46], with episodes of negative inversion [43,47–49].

The study area belongs to the outer Umbria–Marche Apennines chain. The general tectonic–sedimentary evolution of the Umbria–Marche sequence can be framed in three main stages: pre-, syn-, and post-orogenic sedimentation [23]. The pre-orogenic sedimentation is characterized by basin carbonates and marly lithostratigraphic units (Late Triassic to Paleogene in age; [23,35] and references therein, Figure 1). Both syn- and post-orogenic sedimentation is characterized by prevalent terrigenous deposits from Neogene to Quaternary in age and hosted in a wide foredeep basin (Periadriatic Foredeep; [50]). This basin was generated by the flexure of the Adria plate under the Apennine Chain [51], migrating eastward. The foredeep filling includes siliciclastic turbidites (e.g., the Messinian Laga Basin), Plio–Pleistocene marine deposits [51–53], and wedge-top basin sediments [31]. The foredeep itself was gradually involved into the fold-and-thrust belt during the Late Miocene to present.

In the present study, we investigate an area lying in the outer portion of the southern Marche Apennines between the Conero promontory and S. Benedetto del Tronto (Figure 1). In particular, the double effect of the Sibillini thrust to the west and the Gran Sasso thrust to the south (the Abruzzo area in Figure 1) influences the Messinian foredeep geometry and depth. The foredeep hosts thick, internally deformed, turbiditic fan complexes (the Laga Formation; [30,54]) and some positive structures (Acquasanta, Montagna dei Fiori and Coastal Structure) described in the literature (see [2,4,7,30]). The outcropping succession consists of Messinian turbiditic deposits (Figure 2), including a thick, arenaceous basal member whose source is partially provided by the Eocene–Oligocene westernmost chain [7,54] and shallow water facies (S. Donato and Argille a Colombacci Formations). The Argille a Colombacci Formation is always above S. Donato Formation, while the latter may rest discordantly above different members of the Laga Formation (see [22] and references therein). Messinian deposits are followed by the Pliocene succession, whose base marks the marine transgression that occurred after the "lago-mare" phase (sedimentation breck-off; [55]) and the subsequent filling of the Central Adriatic foredeep [56].

The Plio–Pleistocene foredeep basin is associated with deep marine to alluvial sedimentation that shows progressive infill of the basin and a final vertical regressive trend [41]. The infill mainly consists of hemipelagic mudstones deeply incised by coarse-grained canyon-fill deposits [57,58] indicating slope degradation and sediment supply from the uplifted Apennines [32]. Many authors associate these deposits with the outermost part of the orogenic wedge [28,32]; with the formation of thrust fronts and folded structures in the Early Pliocene [11,52]; followed by intense deep water clayey sedimentation in the deepest areas until the Pleistocene; and a new compressive phase right after, likely linked to the reactivation of Late Pliocene thrusts [10]. Deformation of the foredeep via thrusting likely yielded open piggy-back basins and structural highs filled up by shallow-water deposits, likely due to the tightness and blockage of the system against a stable platform, as hypothesized in [11]. The sedimentation within the basin was also partially controlled by the Pliocene–Pleistocene obliquity/precession cycles of the Earth's orbit driving climatic changes, as suggested in [28].

In its outermost portion, the belt shows compressive to transpressive flower structures, which are NW–SE oriented and generally covered by Plio–Pleistocene deposits or partial outcropping on the seafloor. These structures were identified and described in [25] and are located further north of the study area as well as some NE-SW trending faults, which affects the continuity of the previous structures.

In the considered area, the main structural element is the NNW–SSE trending Coastal Structure ("Struttura Costiera" in [11]) which is located near the coastline. This structure continues southwards in the Abruzzo area with similar characteristics [50].

Two main deformation events in the area were recognized by previous authors: an extensional Messinian–Early Pliocene event due to the Adria plate flexure [50] followed by a compressive phase ascribable to the late Early Pliocene [55] or to the Middle Pliocene [50].

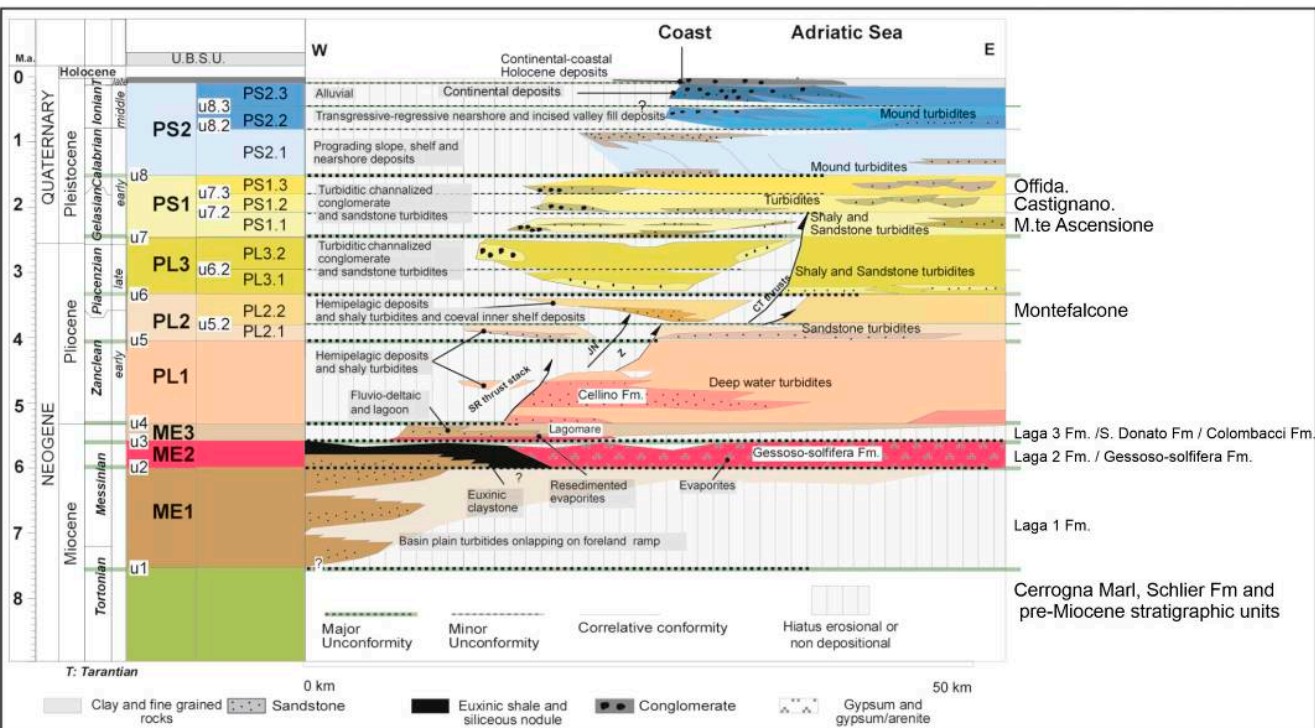

**Figure 2.** Synthetic stratigraphic scheme of the Messinian–Pleistocene of the Central Periadriatic Basin (CPB; slightly modified from [32]). This scheme includes the stratigraphic schemes of previous studies [11,13,27,50,59], but in the right column we list only the units and members of our study area.

## 3. Dataset and Working Methods

In the onshore and offshore areas between the Conero promontory and San Benedetto del Tronto locality, numerous ENI S.P.A. seismic profiles have been interpreted in addition to those available from VIDEPI (https://www.videpi.com/videpi/sismica/sismica.asp; accessed on 30 March 2021). The ENI seismic profiles were migrated, while the VIDEPI ones were stacked and already interpreted. Some ENI profiles were of a MERGE type and good quality, resulting from advanced reprocessing. All the wells available in the area corresponding to the interpreted seismic profiles were used for the interpretation. However, the materials supplied by ENI are confidential, and we are thus not able to represent them on the seismic profiles. Only a general well location was reported. For the seismic velocities of the sedimentary sequences, we referred to [20,60–62]. We then compared the seismic stratigraphy of the seismic profile VIDEPI B-441 001 (Figure 1) with the log data of the Elisa 1 well placed on it (Figure 3). This comparison indicates that velocity, Vp, for the Plio–Pleistocene sequence is 2000 m/s in agreement with the literature in the same area [50,63].

Seismic profiles were then homogenized and scaled to 1:100.000 horizontally and 1 s TWT = 2 cm vertically. In this way, the horizontal and vertical scales were harmonized for the Plio–Pleistocene sequence of the seismic profile, and the geometries of the tectonic and seismic–stratigraphic elements were preserved. As velocity increased at depths below the lower Pliocene deposits, the dip angles of these structures became higher.

To determine the Plio–Quaternary's tectono–stratigraphic evolution, specific seismo–stratigraphic horizons were considered, as follows:

- Top of the Messinian/Pre-Pliocene;
- Near the top of the Early Pliocene;
- Near the base of the Quaternary;
- Unconformities.

Within the interpreted profiles, the seismo–stratigraphic horizons are highlighted with different colours (see Figures 3 and 4 and Plate 1 in Supplementary materials). Unconformities are shown in green dots (see Figures 4, 5, and 7). Some additional reflectors are also highlighted (light blue) because these reflectors allow the main structures to be better marked and identified. The boundary between the Pliocene and Quaternary deposits has been always defined based on the available well stratigraphy, where Calabrian is considered to be the base of the Quaternary, while the new bio–stratigraphic scale from https://stratigraphy.org/ (accessed on 30 March 2021) includes the Gelasian (2.58–1.8 Ma) to Quaternary. This scale could introduce some differences compared to recent cartography [22] but is consistent with [7,62].

Some of the best seismic profiles were selected and organized in 5 almost-parallel Transects with a SW–NE direction within the above-mentioned area. Each Transect is composed of several seismic profiles that are aligned or partially overlapping and aim at realizing a single element.

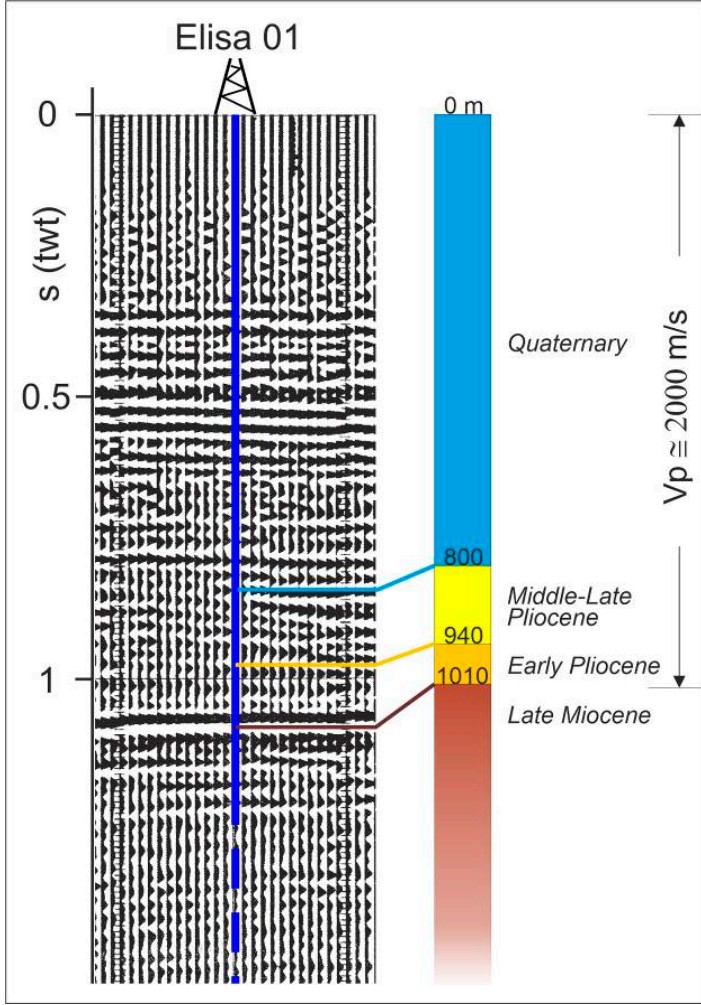

**Figure 3.** Stratigraphic correlation between the ELISA 1 well and a segment of the ViDEPI B-441 seismic profile where the well is placed (Figure 1).

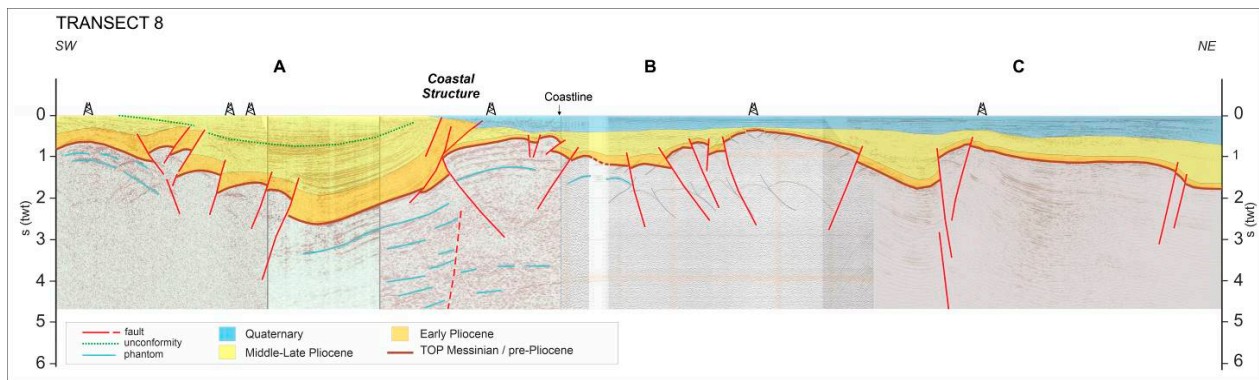

**Figure 4.** Transect 8. The Transect is composed of several seismic profiles labelled with letters *A*, *B*, *C*, see Figure 1). Quaternary deposits are highlighted in blue, Middle–Upper Pliocene deposits are in yellow, Lower Pliocene deposits are in orange, and top Messinian/Pre-Pliocene deposits are in brown. No colour is used for the pre-Pliocene sequence. Unconformities are shown in green dots. Light blue: undefined seismic reflectors. This legend is valid for all Transects (Figures 5–8).

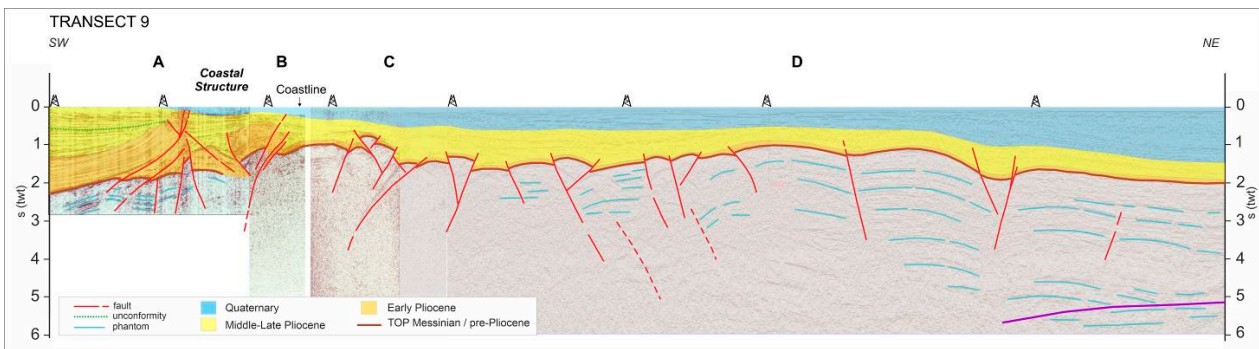

**Figure 5.** Transect 9. The purple line shows the hypothetical basement.

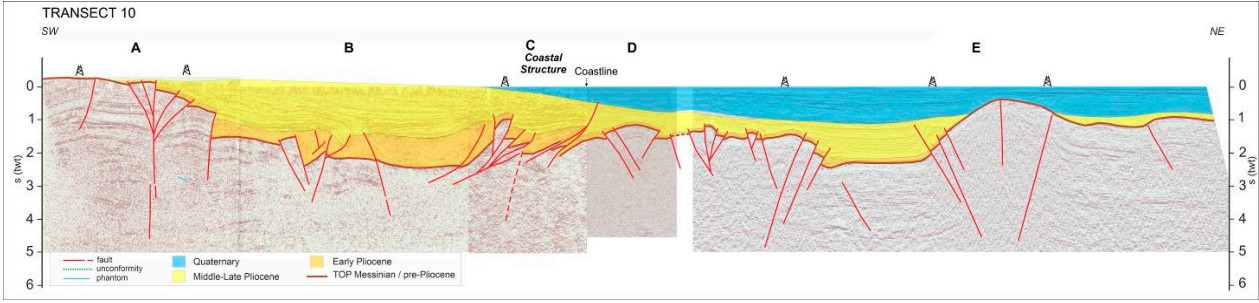

**Figure 6.** Transect 10.

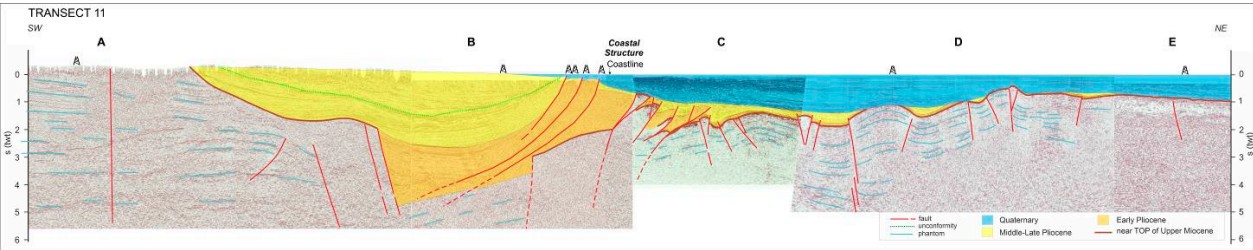

**Figure 7.** Transect 11.

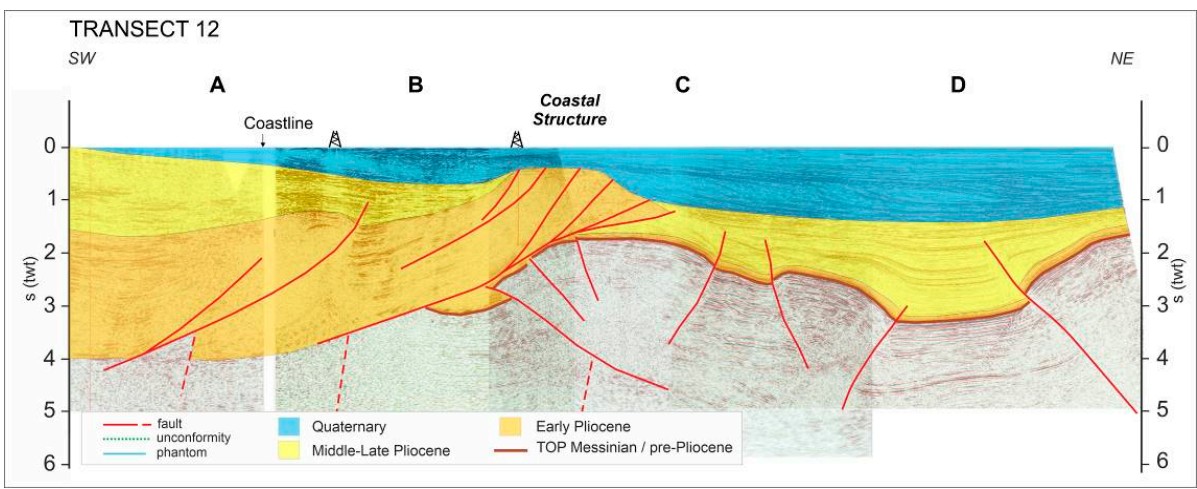

**Figure 8.** Transect 12.

Figure 1 presents traces of the Transects and each seismic profile within them. These traces complete our previous analyses of the outer Apennine Marche sector north of the Conero promontory, where seven Transects have already been observed [24,25]. For this reason, the new Transects are numbered from 8 to 12.

These Transects are represented individually in Figures 4–8 and are reported using a high-resolution plate in the supplementary material (Plate 1).

## 4. Results by Wells and Seismic Profiles Interpretation

### 4.1. Transects

The northernmost Transect (number 8 in this work; Figure 1) developed in the onshore and offshore areas just south of the Conero promontory and includes the seismic reflection profiles *A*, *B*, and *C* (Figure 4). Overall, the quality of these seismic reflection profiles is good; among these profiles, profile *B*, which was already interpreted in the VIDEPI catalogue, was further interpreted here using different colours.

In the onshore area, a WSW–ENE seismic profile (*A*) extends from the east of the Appignano locality to the coastline (ENE). Along this profile, some wells (Figure 1) allow good calibration of the top of the Messinian/Pre-Pliocene and near the top of the Early Pliocene seismo–stratigraphic horizons. An unconformity is also present within the Middle–Upper Pliocene deposits. In the offshore area, two seismic reflection profiles are present (*B* and *C*) and are aligned along a WSW–ENE direction. In particular, the *B* profile partly overlaps the *A* profile, and two hydrocarbon wells occur nearby.

Transect 8 is characterized by three structurally well-defined areas. On the western area a wide syncline is present, affecting a large thickness of about 3000 m Pliocene deposits. Lower Pliocene deposits cover the Messinian deposits in transgression. These deposits have an almost constant thickness, while those above the Middle–Upper Pliocene feature have variable thickness (ranging between 1800 m in the syncline core and about 1000 m along the limbs). Quaternary deposits have a thickness of a few hundred meters.

The western limb of the syncline is affected by a high-angle, W-dipping reverse fault system. This system deforms the whole Lower to Middle Pliocene sequence without involving that of the Upper Pliocene. However, in the westernmost part, some faults deform the overlying unconformity placed within the uppermost part of the Middle–Upper Pliocene sequence.

In the central area, a very complex compressive and uplifted structure ("Coastal Structure") is present. This structure is characterized by shallow East-verging thrusts affecting the Lower–Middle Pliocene sequence and, marginally, the Upper Miocene sequence. Below this structure, an E-dipping reverse fault and a slightly W-dipping sub-vertical fault reaching the relevant depths (>4 s TWT) are present. Quaternary deposits were likely

involved in the deformation of the upper and frontal sectors of this Coastal Structure. As highlighted in the *A* seismic profile, Quaternary deposits outcropping on the western side of the Coastal Structure show reduced thickness compared to those on the eastern side. The Lower Pliocene deposits are indeed less than 100 m in thickness in front of the Coastal Structure and about 1000 m within the syncline behind.

On the eastern area (seismic profiles *B* and *C*), numerous reverse high angle W- and E-dipping faults are present. Overall, the vertical displacement of these faults is moderate, rarely exceeding 500 m. Lower Pliocene deposits have an average thickness of about 100 m or can be absent in the proximity of some structural highs (see *C* in Figure 4 and the wells presented here). The Middle–Upper Pliocene deposits show more variable thickness, from about one hundred meters on the structural highs to more than 1000 m in the proximity of faults and in structurally deeper areas. This extreme variability together with the characteristics of unevenness and chaos of the seismic horizons suggest a syn–tectonic origin of these deposits. The middle lower part of this sequence is certainly affected by reverse faults, while the upper part does not appear to be involved in deformation (seismic profiles *B* and *C* in Figure 4). Indeed, in this area the Quaternary deposits show a regular trend—increasing their thickness toward the east—and are not involved in deformation.

Transect 9 (Figure 5), which includes several wells, shows similar structural and stratigraphic characteristics to those of Transect 8. These differences relate to the greater thickness of the Lower Pliocene and Quaternary deposits facing the Coastal Structure and the high angle faults that are more evident below this structure. A Middle-Upper Pliocene unconformity is also clear in this area and was displaced by frontal thrusts. In this Transect, seismic reflection profile *A* overlaps profile *B* moving eastward toward the coastline. This profile exhibits a shallow compressive structure characterized by east- and west-verging thrusts.

In this structure, the Lower Pliocene deposits are concordant with the Messinian ones, featuring a thickness of about 800 m and more than 1000 m. Eastward, the thickness is notably reduced (about 150 m). Moreover, an unconformity present in the Middle-Upper Pliocene deposits separates the upper portion of the sequence, which is characterized by onlap geometry, from the lower one featuring pinch-out geometry. Furthermore, below the surface thrusts of the Coastal Structure, seismic profiles *A* and *B* from Figures 4 and 5 show very evident high angle W- and E-dipping faults. Offshore, seismic reflection profiles *C* and *D* show pre-Pliocene bedrock widely deformed by high angle west- and east-dipping compressive faults forming gentle pop-up structures with reduced vertical displacement. The thickness of the Lower Pliocene deposits is always very low (<100 m, as also reported in well stratigraphy), while the Middle–Upper Pliocene deposits are syn–tectonic with high variable thickness (from a few to several hundreds of meters) close to compressive structures. Quaternary deposits have a relatively constant thickness (about 600–800 m) and are not affected by deformation. All the other Transects (10–12, see Figures 6–8) show similar characteristics to those described above. As already mentioned, due to the different resolutions of seismic profiles and/or local factors, certain features are clearer than others.

In the westernmost sector of Transects 10 and 11, a deeply rooted sub-vertical structure is highlighted. Transect 10 (Figure 6) shows a branched flower structure that separates the Laga Formation units by the Colombacci Formation at the surface (Figure 1; [25]). This is a branched structure with a possible strike–slip component. In these two Transects, the compressive, W-dipping structures observed in Transect 8 are absent. Furthermore, along Transect 11 (profile *A* in Figure 7), an important normal E-dipping fault (more than 3000 m of vertical displacement) defines the Lower Pliocene basin to the west and is covered by transgressive deposits of the Middle–Upper Pliocene. In the same Transect, the above-mentioned unconformity within the Middle–Upper Pliocene sequence is clearly visible within the syncline. Transect 11 shows that during the Middle/Upper Pliocene, there was simultaneous subsidence (with transgression) in the current onshore to the west together with compression and uplift to the east (Coastal Structure, profiles *A-B* in Figure 7). In both

Transect 11 and 12 (Figure 8), compressive E-dipping faults under the Coastal Structure thrusts are clearly present, as in Transects 8 and 9.

Some thrusts of the Coastal Structure, as shown in Transect 11, affect the Quaternary deposits, such as in Transect 8. In Transect 12, only the shallowest Quaternary deposits are transgressive and are not involved in the deformation. Instead, in Transect 10 the thrusts affect only the Middle–Upper Pliocene sequence. In all Transects, the Quaternary succession covering the offshore flower structure is undeformed. Furthermore, evidence of fore-set Quaternary sedimentation is present in Transects 10 and 12 (Figures 6 and 8).

### 4.2. Characteristics and Distribution of the Plio–Quaternary Deformation

#### 4.2.1. Early Pliocene

Based on well data logs and interpretations of both VIDEPI and ENI seismic profiles, we achieved a reconstruction of the thicknesses and distribution of the Lower Pliocene stratigraphic sequence (Figure 9).

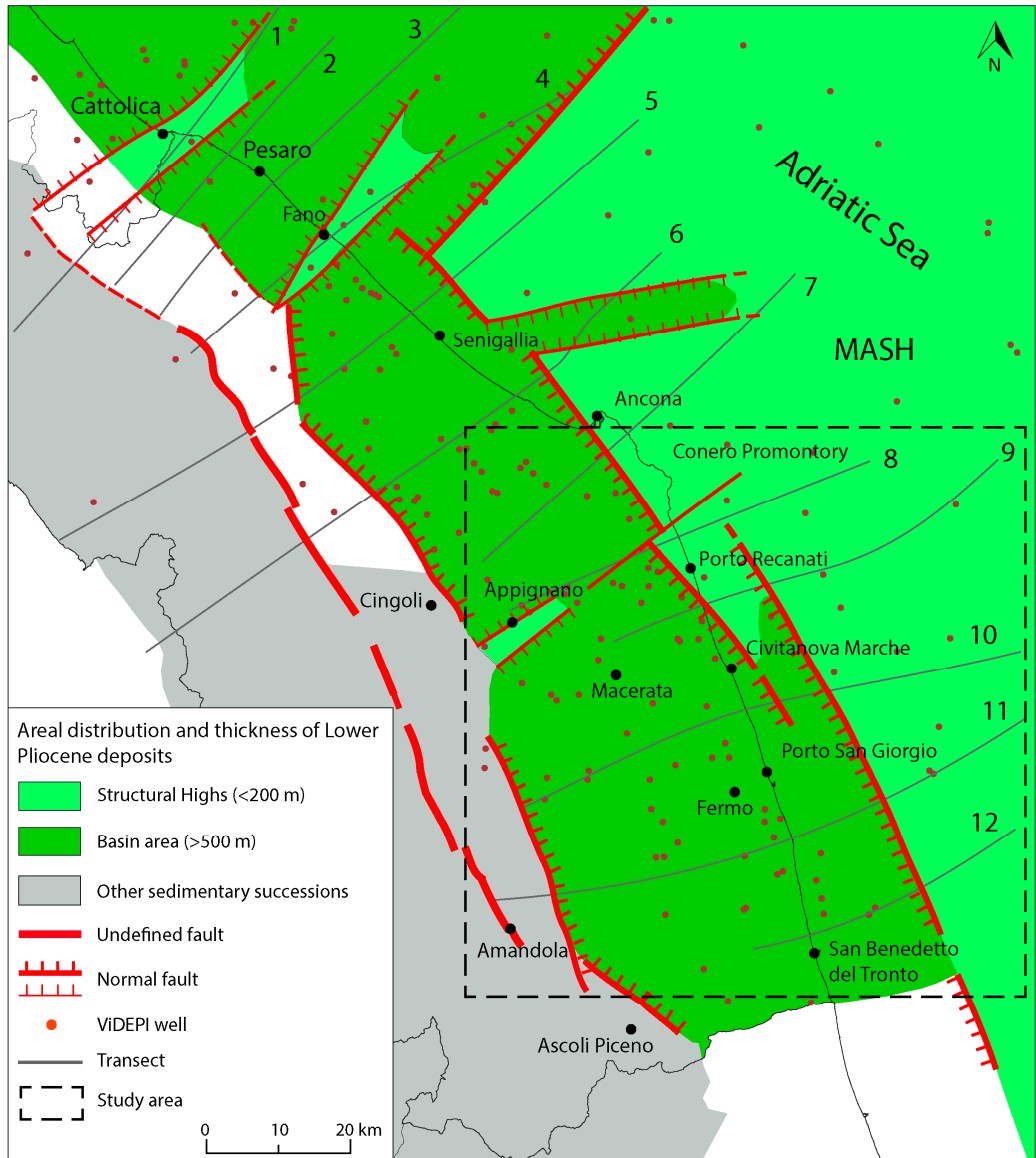

**Figure 9.** Schematic map of the distribution and thickness of Lower Pliocene deposits within the Marche region and the adjacent Adriatic Sea. Marche Adriatic Structural High (MASH) and smaller structural highs and basins are highlighted with the same colour. The study area is located in the dashed square.

This sequence has significant and sudden variations in thickness, and we were able to distinguish between the true sedimentary thicknesses and local tectonic repetitions or highly inclined bedding in proximity of compressive structures. This does not allow us to reconstruct a reliable isopaches map. Afterward, for an immediate view of the Lower Pliocene deposits distribution, we identified two distinct thickness classes (less than 200 m and greater than 500 m). This simple representation makes it possible to easily locate the position, geometry, and kinematics of the faults affecting this sequence. Finally, in Figure 9 the whole outer Marche Apennine sector has been reproduced to show the distribution of this succession.

The thickness distribution of the Lower Pliocene deposits shows evidence of a wide semi-submerged Marche Adriatic Structural High (MASH) area mainly located on the current Adriatic offshore (light green in Figure 9), as well as evidence of a wide basin area located in the northern portion of the same offshore area and in the south onshore area (dark green in Figure 9). This area also includes the northern part of the Marche territory, which is only marginally examined in this work.

Within the MASH area, the thickness of these deposits is very modest (a few tens of meters and, locally, not more than 200 m). In the basin area, the thickness rapidly increases, ranging from more than 500 to 3330 m. The limit between the plateau and basin areas features an NNW–SSE trend south of the Conero promontory lies slightly eastward of the current coastline, which shows instead an NE–SW trend in proximity to the Fano offshore area.

This spatial distribution is likely due to a normal or transtensive fault system that separates the wide and stable MASH area, which appears to be slightly subsident and located in the central–southern Adriatic offshore, from a basin area that is strongly subsident towards its western and northern portions.

Furthermore, the western side of the basin is marked by a normal fault system (Transects 10*A*, 11*A*; Figures 6 and 7). This transtensive fault system was active soon before the onset of the compressive phase highlighted within the Transect.

As underlined in the previous section, this normal fault system consists of syn–sedimentary high angle W- and E-dipping faults characterized by remarkable vertical displacement reaching thousands of meters, which is clearly detectable in the interpreted seismic profiles.

The main faults were likely placed in proximity of the NNW–SSE and NE–SW boundaries of uplifted and subsident areas. Other minor faults further disarticulated both the basin and the MASH areas, defining local thickness variations in the sequence.

### 4.2.2. Middle-Late Pliocene-Quaternary

Based on our investigation, three structurally well-defined areas along a W–E direction are identified (Figure 10).

The western area is characterized by a wide syncline. In the northern part of this area, the syncline is locally intersected by W-dipping high-angle reverse faults (Figure 4); in the southern portion, Lower Pliocene deposits end against a high-angle E-dipping normal fault to the West, covered by transgressive Middle–Upper Pliocene deposits. The syncline axis is about N–S oriented. W of the syncline, a sub-vertical fault system deeply rooted with a N–S trend can be observed.

The central area is marked by a compressive structure (Coastal Structure). This structure consists of a series of E-verging thrusts within the shallower sequence, mainly affecting the Lower–Middle Pliocene deposits and only marginally affecting the Messinian ones. Thrust displacements are rapidly reduced within the Messinian and Lower Pliocene deposits. Just below this horizon, E-dipping reverse faults and deeper high-angle W-dipping faults are present. The Coastal Structure shows an NNW–SSE, almost continuous, trend, and sometimes crops out close to the coastline. This structure was formed starting during the Middle Pliocene, and its deformation continued until the Upper Pliocene, in some parts up to the Quaternary.

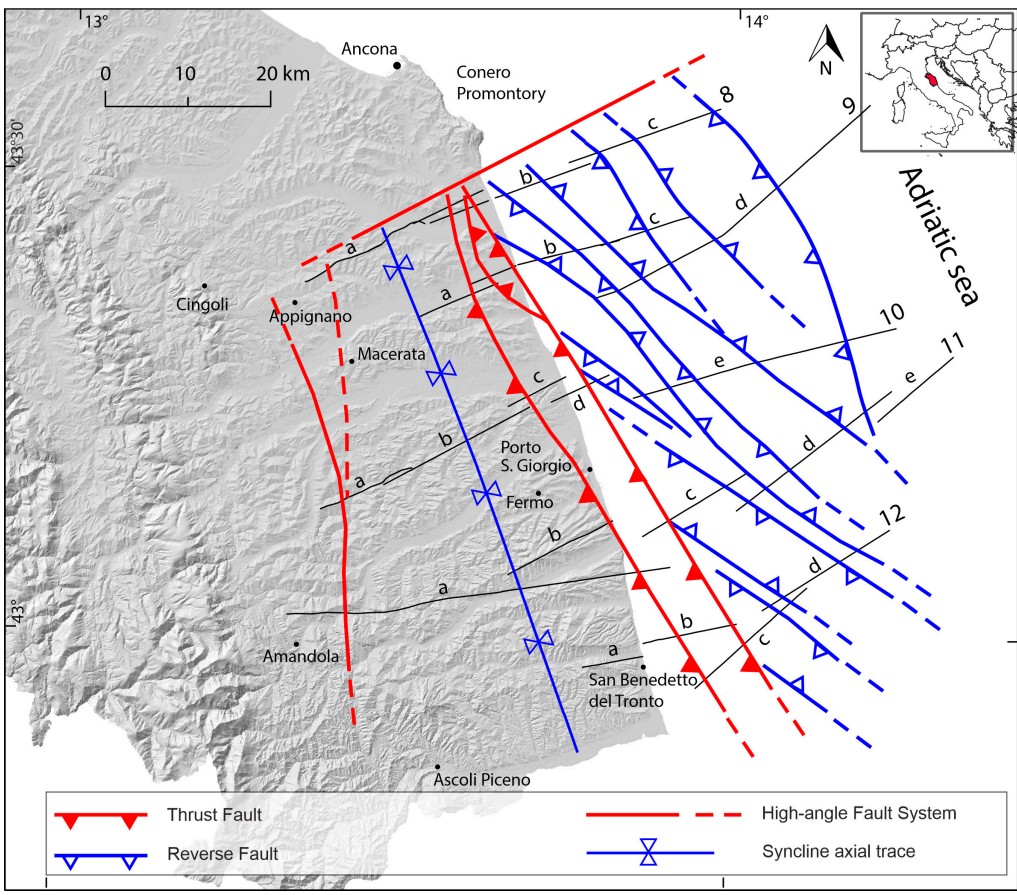

**Figure 10.** Structural sketch map of the outer Marche area south of Conero promontory. The main Plio–Quaternary structures are highlighted.

The eastern area is characterized by W- and E-verging high-angle reverse faults, giving rise to gentle flower structures with an NW–SE trend. These structures were formed from the Middle Pliocene and developed structural highs, some of which were still emerging during the Upper Pliocene/Pleistocene (Transects 10, 11; Figures 6 and 7). Compressive deformation stopped during the Upper Pliocene, and Quaternary deposits were not affected.

## 5. Discussion

During the Messinian, this part of the Marche Apennines outer sector emerged or was close to emersion (the "lago-mare" succession in Figure 2), with sedimentary break-off [55]. The top Messinian/pre-Pliocene seismic horizon was always clearly evident in the examined seismic profiles, with frequent characteristics of an erosive surface (Figures 6 and 7). The Lower Pliocene deposits, however, are often transgressive or discordant over the underlying Messinian or pre-Pliocene ones. Furthermore, no important evidence of Messinian active tectonics was found in this area. This part of the sector started to deform during the Early Pliocene when normal or transtensive faults with an NNW–SSE trend were enucleated. These faults separated heavily subsident basin areas from almost-stable structural highs (Figure 9).

The basin area was mainly located along the current onshore, while the Marche Adriatic Structural High (MASH) was located in the current offshore. This feature continued southward in the Abruzzo region with quite similar characteristics, as described in [28]. According to these authors, in Abruzzo, the basin formed due to horst and graben structures starting in the Messinian–Pliocene transition due to flexural extension of the under-thrusting Adria Plate. In our study area, this extensional phase started in the Early

Pliocene, as indicated by the erosive top Messinian and the transgressive and discordant Pliocene deposits above it.

North of the Conero promontory, the area's slightly more complex setting was also due to an important NE–SW trending fault that segmented the MASH, yielding a basin area to the NW (Figure 9). This structure, already identified in [24], continues from the Fano offshore to the SW along the river valley (Figure 1). Other local features with an NE–SW trend segmented both the basin and the MASH, forming lower-ranking depressions and structural highs (Figure 9). The northernmost transverse structures correspond to the Cattolica seismogenic system [24].

Starting from the Middle Pliocene, a compressive regime was established in the sector south of the Conero promontory, growing the structures underlined in the Transects and in Figure 10.

In more detail, in the study area we highlighted a wide syncline with an almost N–S trend to the west, the Coastal Structure with an NNW–SSE trend in the central portion, and the NW–SE-trending gentle-flower structure system to the east. The syncline was thus formed in correspondence with the Lower Pliocene basin, and the Coastal Structure formed in correspondence with the normal faults bordering the same basin to the east (Figures 9 and 10). The Middle Pliocene deposits are continuous with those of the Lower Pliocene at the syncline core while resting on the same deposits with a pinch-out feature and reduced thickness in proximity to the growing Coastal Structure western flank (Transects 9, 11, and 12; Figures 5, 7 and 8). Variable thickness, with greater thickness close to the faults, attests to the syn–tectonic origins of these deposits in the offshore area.

Further to the west of the syncline, the N–S Amandola-positive flower structure (Figure 6) separates different Messinian units [7]. This structure is high-angle and deeply rooted (Transects 10 and 11; Figures 6 and 7), likely extending farther than the representation in Figure 10. The push-up in the western part of Transect 8a (Figure 4) is likely a continuation of the Amandola structure or one of its branches. All these structural elements are slightly divergent from each other and are interrupted along a complex transverse structure approximately ENE–WSW oriented and located immediately south of the Conero Promontory (Figure 10).

The compressive deformation phase ended in the western and eastern areas during the Late Pliocene-Early Pleistocene. The unconformity within the Middle–Upper Pliocene deposits (Transects 9 and 11; Figures 5 and 7) indicates that the syncline has not deepened since the Late Pliocene. Upper Pliocene deposits rest in an on-lap over the underlying ones above the unconformity and reduce their thickness in proximity of the western flank of the Coastal Structure. These features indicate that, within the syncline, the lower parts of the Middle–Upper Pliocene deposits are syn–tectonic, while those of the upper part are post-tectonic.

The flower structures of the Adriatic offshore are sealed by the Quaternary deposits. In the central area, the Coastal Structure continued its activity even during the Quaternary, as shown in several areas (Transects 8,9, and 11 in Figures 4, 5 and 7). Therefore, all these structures were formed during the Middle Pliocene. Most of these were deactivated at the end of the Late Pliocene, while a few others were probably still active during the Early Pleistocene (Transects 10–12 in Figures 6–8).

The Coastal Structure is characterized by low-angle faults close to the surface and high angle faults at depth. Low-angle faults are mainly involved in the Lower Pliocene deposits, making their repetition clearly visible in all Transects. The underlying Messinian deposits were, instead, not significantly involved, likely due to detachment between the two sequences. In [11], however, Messinian deposits were considered to be strongly involved in deformation. At the western edge of the syncline, and underneath the highly deformed close-to-the-surface succession (Transect 11), parts of the original faults bordering the Lower Pliocene basin are still recognizable. Looking at the Lower Pliocene deposits distribution map (Figure 9), it can be seen that the Coastal Structure is nucleated in the same position as the faults bordering the previous Lower Pliocene basin to the east and

perfectly follows the trend of the latter. Therefore, this structure was formed by partially inverting or deforming (Figure 11) the previous high-angle normal/transtensive faults (see also Figures 5–8). These faults may have acted initially as a barrier, forcing the involved sequences to climb upwards; in some cases (Figures 5, 7 and 8), the innermost thrusts show a higher angle than the external ones. Subsequently, as the shortening increased, the upper parts of the Early Pliocene faults were decapitated (see [43]) and included within the superficial low-angle, E-verging thrust sheets, which mainly affect the Lower Pliocene succession that is partially detached from the underlying one (Figure 11).

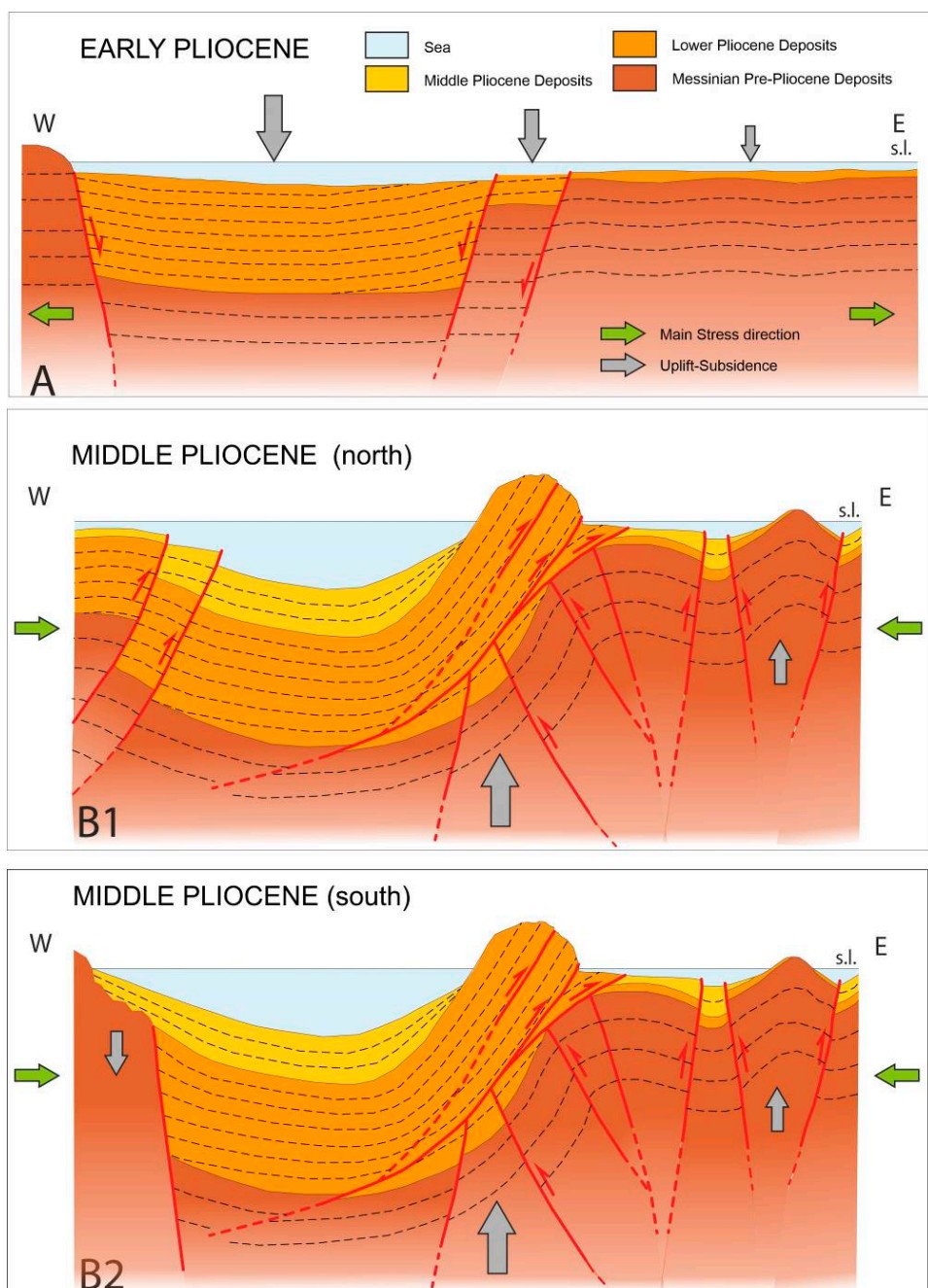

**Figure 11.** Sketch diagram showing the evolution of the Coastal Structure from the Early to Middle Pliocene across two representative cross-sections (not to scale). The size of the grey arrows is proportional to the intensity of vertical movement.

Thus, in the later stage (Late Pliocene–Quaternary), some of the thrust sheets partially covered the westernmost flower structures of the eastern Adriatic area (Transects 8, 10, and 12 in Figures 4, 6 and 8). The compressive Coastal Structure formed due to the inversion of previous extensional features following the "interaction of extensional and contractional deformation" model proposed by [43,64] and in [18,65] for the nearby Montagna dei Fiori and Cingoli structures (Figure 11).

The Coastal Structure continues southward, in the Abruzzo region, with quite similar litho–structural characteristics and ages of deformation [50,65]. Additionally, in that area, E-vergent thrusts are mainly found in the Lower Pliocene deposits, which, in this case, are completely detached from the underlying Messinian ones. However, unlike the process proposed for our study area, these authors suggest that the previous Early Pliocene normal faults bordering the basin were already enucleated during the Messinian. Furthermore, these normal faults were not involved in compressive deformation but were simply covered by the thrust sheets. According to these authors, compressive deformation began in the Early Pliocene.

As a result of the compression that determined the Coastal Structure's development, tilting of the block between this structure and the Amandola structure to the W likely also have occurred. During the Middle Pliocene, there was simultaneous uplift of the eastern front (enucleation and uplift of the Coastal Structure) and subsidence of the western side (transgression of the Middle Pliocene deposits on those previously raised during the pre-Pliocene time; Transect 11). The horizontal rotational axis may correspond to the syncline axis. This mechanism is similar to that described in [66] for the Po Valley. During the Late Pliocene–Quaternary this rotation ceased, and the deposits of the same age became horizontal.

The Amandola structure, the syncline, and the Coastal Structure show a straight and regular trend. As previously mentioned, the trend of these main onshore structures is somewhat divergent from the offshore one, even though they all formed during the same time interval. This can be attributed to the influence of pre-existing features inherited by previous deformation phases such as the faults shown in Figure 9. These structures are compatible with the main local shortening oriented in an NNE–SSW direction during the Middle–Late Pliocene (compression with the P axis about NNE–SSW; Figure 12), which emerged in the northern sector of the Marche region [24,67,68] and, more generally, in the overall Central Adriatic area [69,70].

In this context, right-lateral transpression developed along the Coastal Structure and likely enhanced the gentle flower systems of the Adriatic offshore (Figure 12)

The Coastal Structure schematically represented as continuous and regular in Figure 10 is most likely composed of several structures, some of which were still active during the Quaternary, as shown by fairly significant earthquake sequences ($M_w$ = 5, Porto San Giorgio sequence, [71,72]) that occurred recently (Figure 12).

As previously mentioned, the described structures were somehow interrupted to the north along a transverse ENE–WSW-oriented structure. The existence of transverse faults has been highlighted in literature by various authors, particularly in the Marche-Abruzzo onshore (see [7,24,73] and reference therein). In our study area, several structures underwent sudden changes in characteristics (age of deformation, geometry, and direction) that are observable when compared to those mapped in [24] in the areas west, east, and north of the Conero promontory. Furthermore, the structures present immediately northward of this transverse element and of our study area, e.g., the Early Pliocene transpressive structure of Strada-Moie and S. Andrea di Suasa (see Figure 7 in [23]) are no longer present in the south. Indeed, in this southern area, extension was still occurring during the Early Pliocene.

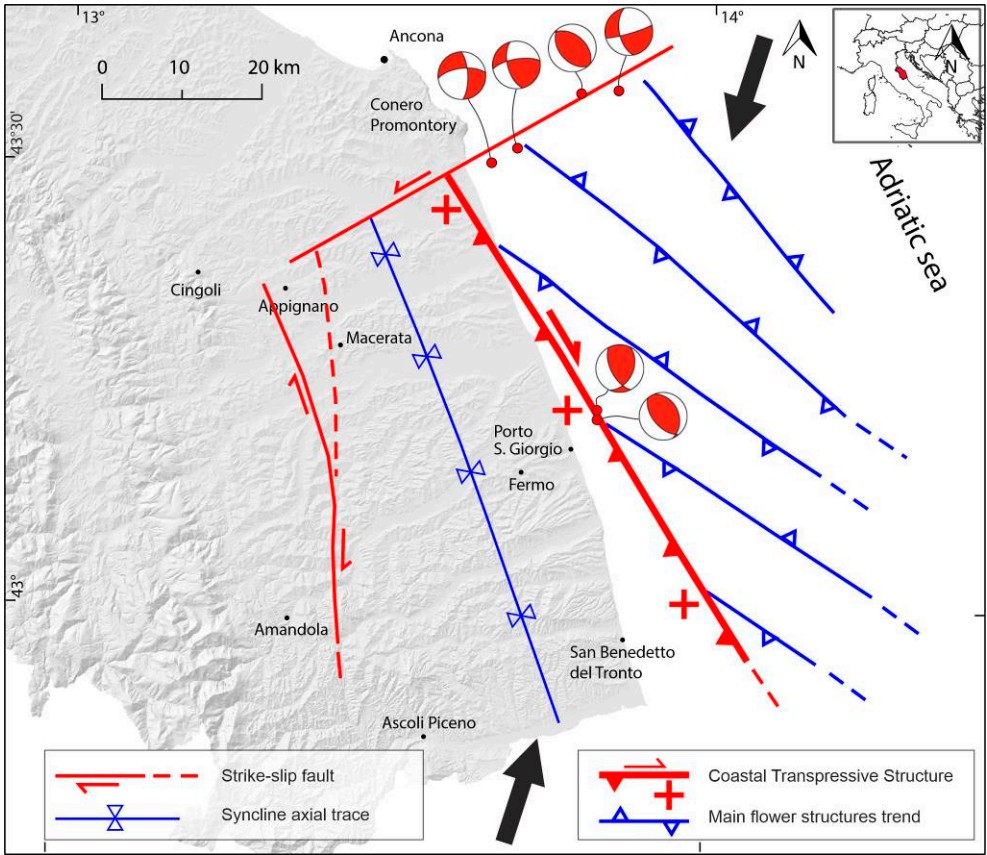

**Figure 12.** Kinematic sketch map. Red lines indicate fault systems still active during the Quaternary. The black arrow represents the main shortening direction, red arrows describe the right lateral strike–slip component, and the plus symbol (+) is the narrow strongly uplifted area. Focal mechanisms (beach balls) of the main earthquakes of 1987 Porto S. Giorgio and 2013 south of Conero seismic sequences are shown.

The transverse structure south of the Conero promontory, already partially present in [7], interrupts structures with Quaternary activity, i.e., the Coastal Structure to the south and the Conero compressional structure [25] to the north. Therefore, this tectonic element must be Quaternary itself, as also attested by recent earthquakes and seismic sequences in the offshore along the element (Figure 12). These focal mechanisms are predominantly strike–slip, with P-axes oriented around the ENE–WSW and sub-vertical planes [67,68]. Furthermore, the epicentres of the seismic sequences described by these authors are aligned ENE–WSW.

## 6. Conclusions

Seismic profile interpretations and well stratigraphic data allowed us to describe the Plio–Quaternary evolution of the outer Marche Apennines south of the Conero promontory. The main results can be summarized as follows:

- During the Early Pliocene, the area was affected by extensional or transtensive tectonics, resulting in the formation of a strongly subsident basin and a more stable structural high. More than 3000 m of sediment accumulated in the basin zone, while the structural high (MASH) hosts less than 200 m of Lower Pliocene deposits.
- The basins and structural highs are separated by an approximately NNW–SSE normal and transtensive fault system located close to the current coastline. Other normal faults with an NNW–SSE trend developed in the current onshore area and border the basin to the W. The structural high is instead located in the current offshore area.
- Starting from the Middle Pliocene, the entire area underwent compression, with the P axis oriented about NNE–SSW leading the formation, from W to E, of the NNW-SSE

dextral strike-slip Amandola structure, the NNW–SSE dextral transpressive Coastal Structure, and an NW–SE-striking system of gentle flower structures (offshore).

- The Coastal Structure is the most complex and important structure in the study area. It consists of an E-vergent thrust system at surface and high-angle E and W-vergent faults at depth. Shallow thrusts mainly affected the Pliocene deposits and, locally, the Quaternary ones. The mainly Messinian underlying deposits were marginally involved in deformation. Deeper faults affect Mio-Pliocene and older deposits. As a result, in the shallower part of the Coastal Structure, pre-existing normal faults were inverted or crosscut and incorporated into the ongoing thrusts, while at depth, they were not deformed.
- The trends of the Coastal Structure and the flower structures within the offshore are slightly divergent despite being contemporaneous because the former was strongly influenced by inherited structures.
- The compressive phase was finished during the Late Pliocene in the syncline, as well as along the flower structures. The Coastal Structure was still active during the Quaternary. This is also testified by recently recorded seismic activity.
- A complex transverse structure with a general ENE–WSW trend (at least partially active and seismogenic) traces the boundary between the outer areas north and south of the Conero promontory, where the styles, geometries and times of deformation of the Plio–Quaternary structures are significatively different.

Based on our results, we conclude that during the Plio–Quaternary times, this portion of the outer Apennine sector is mainly affected by a right-lateral transpressive deformation, and by widespread kinematic inversion of pre-existing structures. Former studies proposed a simple E-vergent compressive deformation for the same area.

**Supplementary Materials:** The following are available online at https://www.mdpi.com/article/10.3390/geosciences11050184/s1.

**Author Contributions:** Investigation, M.C.; resources, J.C., M.C., S.T, and P.P.P.; data curation, S.T., P.P.P., and M.C.; writing—original draft preparation, C.I., M.C., and J.C.; writing—review and editing, M.C., C.I., and J.C.; funding acquisition, P.P.P. and C.I. All authors have read and agreed to the published version of the manuscript.

**Funding:** This research was supported by grants from P.P. Pierantoni (FAR—Fondo di Ricerca Ateneo, Università di Camerino) and C. Invernizzi (Interreg Coastenergy project ID: 10045844; FAR—Fondo di Ricerca Ateneo, Università di Camerino).

**Institutional Review Board Statement:** Not applicable.

**Informed Consent Statement:** Not applicable.

**Acknowledgments:** ENI S.p.A. is thanked for supplying seismic profiles, for their facilities, and for permission to publish the data. Stefano Mazzoli and Claudio Di Celma are kindly thanked for their useful discussions. The authors also thank the reviewers for several insightful comments that consistently improved the manuscript and Giancarlo Molli, Angelo Cipriani, and Domenico Liotta for their editorial work.

**Conflicts of Interest:** The authors declare no conflict of interest.

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
