# Peer review of "Plio–Quaternary Structural Evolution of the Outer Sector of the Marche Apennines South of the Conero Promontory, Italy"

_geosciences, doi:10.3390/geosciences11050184_

Round 1

Reviewer 1 Report

Review Report for Geosciences - Manuscript ID geosciences-1111730

Title: Plio-Quaternary compressive deformation in the outer sector of the Marche Apennines: time space characteristics and distribution South of the Conero Promontory

Authors: Mario Costa , Jessica Chicco , Chiara Invernizzi * , Simone Teloni , Pietro Paolo Pierantoni

22nd February 2021

Comments and Suggestions for Authors

General comment

This paper mainly concerns on the reconstruction of the Plio-Quaternary evolution of a portion of the outer sector of the central Apennines. The geometry of thrust front and foredeep basin derive from the basin-scale mapping of 3 sedimentary sequences, delimited by at least 3 regional unconformities, that are constrained by well data and seismic interpretation. During the Lowe Pliocene, the foredeep basin results to be dominated by east- and west-dipping normal faults accommodating areas with distinct rates of subsidence. Starting from Middle Pliocene, several compressive structures interacted with and locally inverted the pre-existing extensional faults. Finally, during Quaternary times, a right-lateral strike-slip tectonics affected the study area and some of the inherited normal/compressive structures were additionally reactivated.

The methodological approach to reconstruct structural setting of the study area by full integration of subsurface data consisting of confidential and public seismic reflection profiles and well-logs is rigorous and clearly described in the text.

The main results are new and original, and they consist of five transects of interpreted composite seismic profiles, structural maps with simplified isopach of the Plio-Pleistocene succession and mapping of the main tectonic lineaments, and a synoptic sketch of the tectono-sedimentary evolution of Apennines foredeep-outer thrust front. The illustrated dataset supports efficiently the interpretations and conclusions by the Authors; in particular, the manuscript provides a novel reconstruction of the physiography of the central Adriatic foredeep basin and a different interpretation of the along-strike variations in the recent evolution.

Overall, the topic is of prime interest for a wide, international readership and the paper provides new and very interesting seismic data from the east Marche-offshore region of the Adriatic foreland, whose structure and evolution are poorly understood when compared to the evolution of the adjacent Apennine belt onshore. However, the manuscript in the present form suffers in several ways. The main reasons of concern are:

  • from the present introduction and discussion sections, the reader cannot entirely appreciate why, for example, the recognition of pre-thrusting extension in the Lower Pliocene foredeep basin should be relevant for a broad, international audience and which are the implications in the evolution of foredeep basin and subsequent fold-and-thrust belt. It is my personal view that the analytical description of the transects should be preceded by a robust introduction on the architecture of foredeep basins and adjacent foreland domains, and on the importance of the pre-thrust architecture in controlling the location, geometry and evolution of subsequent compressive structures. This introduction, with emphasis on previous work and on the processes that rule the pre-orogenic intraplate deformations of foreland domains, would make the general topic interesting for a wide, international readership. In this broader context, it should be stressed that the Apennines of Italy, and especially the Adriatic foreland domain, represent an example where it is possible to infer a foreland deformation process from the geological and geophysical record and to document an interaction between normal, thrust and strike-slip faults. Moreover, a more general introduction will provide the basis to discuss for a comparison with previously proposed models for the Apennines, and for analogue thrust belts elsewhere.
  • The second paragraph ‘Geological setting’ is not clear and focalized to the main topic of the paper (e.g., several phrases are devoted in describing the Messinian formations that are not interpreted in the subsequent seismic sections or illustrated in the geological map). The text is too generic, sometimes confusing, and very poor in English. This part of the manuscript needs to be clearly rewritten to introduce to a potential international reader to the main structural and stratigraphic rewritten to better illustrate the main structural and stratigraphic elements of the study area within the general context of the Apennines chain-Adriatic foreland; I also suggest to add a general scheme to show the location of the study area within the Apennines fold-and-thrust belt.
  • In general, the English form is quite poor in both grammar and style. The manuscript should have been thoroughly revised by a mother-language scientist, or at least by a reader who is fluent in English, before its final submission.

Specific comments

  • Authors need to indicate how they calibrated the interpreted interval reflectors and which method they adopted for the well-to-seismic tie (e.g., check-shot data, VsP data, synthetic seismograms). A figure including an enlarged part of a seismic profile with the calibration of key horizons/sequences by a well could be very informative.
  • Some chronostratigraphic aspect adopted in the subdivision of the Plio-Quaternary succession need to be more explicitly illustrated in the 3.1 section. In particular, the use of the ‘old’ Pliocene subdivision (tripartite Pliocene) and the Gelasian included il their Late Pliocene.
  • Authors common refers to low- and high-angle faults in their seismic profiles but in their figures they do not indicate any apparent dip (i.e., due to distortion of the vertical scale in twt with respect to the horizontal scale). I would suggest to include scale and vertical exaggeration in ALL figures (Figs. 3-7 and Fig. 10) or to add a further figure with a time-to-depth conversion of an interpreted seismic profile.
  • Authors introduce the term ‘Marche Adriatic Plateau (MAP)’ to indicate the NW-SE trending Pliocene structural high transecting the central Adriatic foreland. However, in my opinion this is not properly a plateau (i.e., a flat area) and this strongly deformed zone within large part of the central Adriatic (not only limited to the Marche) strictly corresponds to the Mid-Adriatic Ridge (e.g., De Alteris, 1995; Scisciani, 2009; Pace et al., 2015; Scisciani et al., 2019). I suggest to discuss about this.
  • The Fig. 9 is too late and is difficult for a reader to follow your descriptions in the section 3 without a reference to a structural map.
  • The Coastal Structure is a regional thrust related fold which is considered by several authors active since the Lower Pliocene (at least during the Gb. puncticulata Biozone) as suggested by stratigraphy of several wells drilled in the crest of the anticline both to the south (e.g., San Benedetto 1, 2 and Tortoreto 1 wells) and north (Camerano 1 and Osimo 1 dir wells) of your study area (e.g., Dattilo et al., 1999; Scisciani & Montefalcone, 2005). Further detailed descriptions of your date and discussion including other interpretations are needed.

Detailed comments to the text, figures and references are included in the annotated version of the submitted manuscript (see the attached .pdf file).

Based on the above statements, the topics and the results are of interest for an international audience, and thus the manuscript is appropriate for being published in Geosciences. Therefore, I recommend that it is published with minor/major (scientific contents/English style, respectively) revisions.

With my very best regards

Author Response

This paper mainly concerns on the reconstruction of the Plio-Quaternary evolution of a portion of the outer sector of the central Apennines. The geometry of thrust front and foredeep basin derive from the basin-scale mapping of 3 sedimentary sequences, delimited by at least 3 regional unconformities, that are constrained by well data and seismic interpretation. During the Lowe Pliocene, the foredeep basin results to be dominated by east- and west-dipping normal faults accommodating areas with distinct rates of subsidence. Starting from Middle Pliocene, several compressive structures interacted with and locally inverted the pre-existing extensional faults. Finally, during Quaternary times, a right-lateral strike-slip tectonics affected the study area and some of the inherited normal/compressive structures were additionally reactivated.

The methodological approach to reconstruct structural setting of the study area by full integration of subsurface data consisting of confidential and public seismic reflection profiles and well-logs is rigorous and clearly described in the text.

The main results are new and original, and they consist of five transects of interpreted composite seismic profiles, structural maps with simplified isopach of the Plio-Pleistocene succession and mapping of the main tectonic lineaments, and a synoptic sketch of the tectono-sedimentary evolution of Apennines foredeep-outer thrust front. The illustrated dataset supports efficiently the interpretations and conclusions by the Authors; in particular, the manuscript provides a novel reconstruction of the physiography of the central Adriatic foredeep basin and a different interpretation of the along-strike variations in the recent evolution.

Overall, the topic is of prime interest for a wide, international readership and the paper provides new and very interesting seismic data from the east Marche-offshore region of the Adriatic foreland, whose structure and evolution are poorly understood when compared to the evolution of the adjacent Apennine belt onshore. However, the manuscript in the present form suffers in several ways. The main reasons of concern are:

  • from the present introduction and discussion sections, the reader cannot entirely appreciate why, for example, the recognition of pre-thrusting extension in the Lower Pliocene foredeep basin should be relevant for a broad, international audience and which are the implications in the evolution of foredeep basin and subsequent fold-and-thrust belt. It is my personal view that the analytical description of the transects should be preceded by a robust introduction on the architecture of foredeep basins and adjacent foreland domains, and on the importance of the pre-thrust architecture in controlling the location, geometry and evolution of subsequent compressive structures. This introduction, with emphasis on previous work and on the processes that rule the pre-orogenic intraplate deformations of foreland domains, would make the general topic interesting for a wide, international readership. In this broader context, it should be stressed that the Apennines of Italy, and especially the Adriatic foreland domain, represent an example where it is possible to infer a foreland deformation process from the geological and geophysical record and to document an interaction between normal, thrust and strike-slip faults. Moreover, a more general introduction will provide the basis to discuss for a comparison with previously proposed models for the Apennines, and for analogue thrust belts elsewhere.
    • We thank the Reviewer for his comment. In the revised text we reorganized both the Introduction and Geological setting. Now, in the Introduction a contextualization of general problems relative to foredeep architecture and evolution, and the influence of inherited structured (pre-thrust architecture) are well addressed. To favor a wider interest, more references and connection to other areas are presented.
  • The second paragraph ‘Geological setting’ is not clear and focalized to the main topic of the paper (e.g., several phrases are devoted in describing the Messinian formations that are not interpreted in the subsequent seismic sections or illustrated in the geological map). The text is too generic, sometimes confusing, and very poor in English. This part of the manuscript needs to be clearly rewritten to introduce to a potential international reader to the main structural and stratigraphic rewritten to better illustrate the main structural and stratigraphic elements of the study area within the general context of the Apennines chain-Adriatic foreland; I also suggest to add a general scheme to show the location of the study area within the Apennines fold-and-thrust belt.
    • The Geological setting has been completely rewritten in order to better focalize to the main topic of the paper and to introduce an international reader to the area. We believe that now the main structural and stratigraphic elements of the study area within the general context of the Apennines chain-Adriatic foreland are better defined also with several additional references and a view on the model evolution for this area though time.
  • In general, the English form is quite poor in both grammar and style. The manuscript should have been thoroughly revised by a mother-language scientist, or at least by a reader who is fluent in English, before its final submission.
    • We have provided for the revision of the text by a mother-language

Specific comments

  • Authors need to indicate how they calibrated the interpreted interval reflectors and which method they adopted for the well-to-seismic tie (e.g., check-shot data, VsP data, synthetic seismograms). A figure including an enlarged part of a seismic profile with the calibration of key horizons/sequences by a well could be very informative.
    • A specific paragraph was introduced with these informations and an extra figure (now Fig. 3) including an enlarged part of a seismic profile (by Videpi) and the calibration of key horizons/sequences from a well has been introduced
  • Some chronostratigraphic aspect adopted in the subdivision of the Plio-Quaternary succession need to be more explicitly illustrated in the 3.1 section. In particular, the use of the ‘old’ Pliocene subdivision (tripartite Pliocene) and the Gelasian included il their Late Pliocene.
    • We better specified the use of tripartite Pliocene stratigraphy both in the Abstract and in 3.1 paragraph with an additional sentence
  • Authors common refers to low- and high-angle faults in their seismic profiles but in their figures they do not indicate any apparent dip (i.e., due to distortion of the vertical scale in twt with respect to the horizontal scale). I would suggest to include scale and vertical exaggeration in ALL figures (Figs. 3-7 and Fig. 10) or to add a further figure with a time-to-depth conversion of an interpreted seismic profile.
    • We better specified about this aspect in 3.1 paragraph, referring to velocities and we also added a further figure (Figure 3 in the revised text)
  • Authors introduce the term ‘Marche Adriatic Plateau (MAP)’ to indicate the NW-SE trending Pliocene structural high transecting the central Adriatic foreland. However, in my opinion this is not properly a plateau (i.e., a flat area) and this strongly deformed zone within large part of the central Adriatic (not only limited to the Marche) strictly corresponds to the Mid-Adriatic Ridge (e.g., De Alteris, 1995; Scisciani, 2009; Pace et al., 2015; Scisciani et al., 2019). I suggest to discuss about this.
    • We changed this name to “Marche Adriatic Structural High” (MASH) considering the reviewer’s observation
  • The Fig. 9 is too late and is difficult for a reader to follow your descriptions in the section 3 without a reference to a structural map.
    • We anticipated this figure in the revised text
  • The Coastal Structure is a regional thrust related fold which is considered by several authors active since the Lower Pliocene (at least during the Gb. puncticulata Biozone) as suggested by stratigraphy of several wells drilled in the crest of the anticline both to the south (e.g., San Benedetto 1, 2 and Tortoreto 1 wells) and north (Camerano 1 and Osimo 1 dir wells) of your study area (e.g., Dattilo et al., 1999; Scisciani & Montefalcone, 2005). Further detailed descriptions of your date and discussion including other interpretations are needed.
    • We believe we have justified in the revised text the reasons that led us to attribute the beginning of the compression phase to the Middle Pliocene, especially in the Coastal Structure. Carruba et al. (2006) refer that in the southward continuation (Abruzzo) of the Marche Coastal Structure it began in the Middle Pliocene. However, we have also reported the different opinions of other authors (De Alteriis, 1995). In our area the the Lower Pliocene deposits are so intensely deformed (transepts 8-12) within the Coastal Structure that they do not allow to establish with certainty the beginning of the compressive deformation and the wells drilled in correspondence with this structure, fall in the areas of most intense and complex deformation caused by overlapping and repetition of thrusts sheets.

Detailed comments to the text, figures and references are included in the annotated version of the submitted manuscript (see the attached .pdf file).

            We answered to the detailed comments point by point in the Reviewer’s annotated version of the pdf

Reviewer 2 Report

Dear Authors,

thanks for your manuscript, which deals with the subsurface information from the southern part of the Marche thrust front also on shore. You show several cross-sections and maps of the area related to the middle Pliocene - earliest Quaternary tectonics and an attempt of sketched restoration. You also present a reconstruction of the Early Pliocene pre-thrusting normal faults affecting the Marche region. In the discussion this evidence allows you to contextualize the Coastal structure as a transpressive feature of the Apennine thrust front. I find very promising the potentiality of applicability of the examples that you have provided on the interaction between inherited extensional structures and the subsequent compressive structures that the paper deals with.

In my opinion, figures need legends and captions are too short or not existing. Most figures need to be re-edited according to the journal standards. Their order shall be reviewed following the suggestions in the reviewed PDF attached. In particular, the geological cross-sections need to be possibly put in one panel to finally provide a clue of the subsurface interpretation in support of the reactivation of pre-existing faults and/or of the interaction between relatively deeply seated and shallower faults accommodating shortening. In order to get the paper of more general interest, the literature study should expand to similar works in the region and elsewhere. The work brings new information on the area and provides an interesting relationship between the observed seismicity in the region, which is not supported graphically nor, unfortunately, is well-presented in the description but at the end of the discussion.

I suggest to focus the introduction, providing a more clear contextualization in the thematics of the timing and mode of development of the Apennine thrust front during Neogene and Quaternary. I also invite you about reconsidering the presentation of the methods and the data, which I suggest to occur in two separated sections (i.e., methods, results). The borehole data need to be better associated with their seismic lines so I have suggested to tight a seismic log with a reference well.

The discussion suffers of the same problem of the introduction as it deals with a very limited numbers of cited works, all of regional interest.

In the discussion, when you establish the link between the observed structures and the literature, the interpretation of the data should be more anchored to what was shown in their figures.

I have found some inconsistencies and I have carefully reviewed both text and figures, providing many suggestions of change that I expect will be useful to you.

At its current stage, I find this work not yet suitable for the public of Geosciences but after a major review of both text and figures, it should be fine. 

Best wishes,

Author Response

Dear Authors,

thanks for your manuscript, which deals with the subsurface information from the southern part of the Marche thrust front also on shore. You show several cross-sections and maps of the area related to the middle Pliocene - earliest Quaternary tectonics and an attempt of sketched restoration. You also present a reconstruction of the Early Pliocene pre-thrusting normal faults affecting the Marche region. In the discussion this evidence allows you to contextualize the Coastal structure as a transpressive feature of the Apennine thrust front. I find very promising the potentiality of applicability of the examples that you have provided on the interaction between inherited extensional structures and the subsequent compressive structures that the paper deals with.

In my opinion, figures need legends and captions are too short or not existing. Most figures need to be re-edited according to the journal standards. Their order shall be reviewed following the suggestions in the reviewed PDF attached. In particular, the geological cross-sections need to be possibly put in one panel to finally provide a clue of the subsurface interpretation in support of the reactivation of pre-existing faults and/or of the interaction between relatively deeply seated and shallower faults accommodating shortening.

We now present a revised version of the caption to figure, with more details, and following the suggestion in the attached pdf. We also add a panel (Plate 1) in the supplementary material where the sections are presented all together at large scale to facilitate the reading or the structures in a 3D way

In order to get the paper of more general interest, the literature study should expand to similar works in the region and elsewhere. The work brings new information on the area and provides an interesting relationship between the observed seismicity in the region, which is not supported graphically nor, unfortunately, is well-presented in the description but at the end of the discussion.

            In this respect, we expanded the cited literature and the examples taken form other regions to support our findings and descriptions. We also added in the last figure a graphical support for the seismicity presented in the description

I suggest to focus the introduction, providing a more clear contextualization in the thematics of the timing and mode of development of the Apennine thrust front during Neogene and Quaternary.

            We modified the Introduction and the revised text now provides a more clear contextualization of the problems and of the study area

I also invite you about reconsidering the presentation of the methods and the data, which I suggest to occur in two separated sections (i.e., methods, results). The borehole data need to be better associated with their seismic lines so I have suggested to tight a seismic log with a reference well.

            We accepted this advice introducing two separated paragraphs: 3.1. Seismic reflection profiles and well data (methods) and 3.2  Seismic profiles interpretation (data interpretation). We also tighted a seismic log with a reference well in the new Figure 3.

The discussion suffers of the same problem of the introduction as it deals with a very limited numbers of cited works, all of regional interest.

            We implemented the cited works also in the discussion to widen the discussion itself and we believe that this piece of work in now better organized

In the discussion, when you establish the link between the observed structures and the literature, the interpretation of the data should be more anchored to what was shown in their figures.

            We worked on this.

I have found some inconsistencies and I have carefully reviewed both text and figures, providing many suggestions of change that I expect will be useful to you.

            Thank you. We carefully considered the numerous suggestions

At its current stage, I find this work not yet suitable for the public of Geosciences but after a major review of both text and figures, it should be fine. 

            We now believe that the new text file and revised figures (with an additional figure too) are in a suitable form

Round 2

Reviewer 2 Report

Dear Authors,

thanks for your second version of manuscript, which deals with the subsurface information from the onshore to offshore southern part of the Marche Apennines. I recognize that by following part of the suggested changes, you have implemented the text, especially in the geological setting. The topic is of great interest to anyone who deals with fault reactivation and foreland basins. You also made some progress concerning the description of the cross-sections and maps. However, in my opinion there are still major changes to the structure of the paper that will help refining methods, results and discussion in order to get to a sound geological model of the area related to the middle Pliocene - earliest Quaternary tectonics. Importantly, the paper has to present a take home message that has to deal with the existing literature on a regional scale. Further, as it deals with reactivation of relatively recent inherited structures, it has to be based on a strong syn-sedimentary evidence.

The suggestions provided in the first review round, where partially skipped. As a result, figures still need to be re-edited not just to follow the journal standards but also to reach a geological description that can support the aimed interpretation regarding the recent structural evolution of the area. I have to remark that their order shall be reviewed following the suggestions in the second reviewed PDF attached.

With pleasure, I have dedicated care again into commenting again throughout the whole paper. I hope that this effort can help you highlighting the results of your research. You replied that you have accepted that comment but I still see the initial structure which is not the classical Introduction, Geological setting, Methods and materials, Results, Discussion, which helps also reading the paper. Beside the structure of the paper, that is presented with an unusual mixture/transition of methods and results with interpretation maps into the results, I was concerned about the change of the structural interpretation model among the different geological transects presented, for example regarding the geometry of the thrusts.

ABSTRACT

I would focus the beginning of this abstract on the structural problem you aim to show: positive vs negative reactivation of inherited structures in fold and thrust belts or in the Apennines. Here you present an example from the external domain of the Marche Apennines, which display interesting reactivation examples from the subsurface geology explored.

INTRODUCTION
The Introduction needs to be a little more improved. In this frame, beside the local interest of the southern Marche structures, you should contextualize your aim into the foreland tectonics of the Apennines and try to make a general outcome out of your results (e.g., on the reactivation or deformation of preexisting structures). Those can be your guidelines in the discussion too.

Another point you can better exploit is regarding the presentation of the previous Author work. I have appreciated that you have increased the number of citations. However, they need to be explained and contextualized. Numbering them in loads is not suitable as it does not help making distinction between the different contributions from the different authors. This is relevant as it helps focusing the introduction to the problem that you want to solve and it later helps guiding the discussion towards a more cutting-edge analysis of the regional structures. Have a look at the comments for details.

GEOLOGICAL SETTING

The structure of the geological setting is unusual too. As focusing into a geological introduction goes from a larger point of view to a more restricted area of observation, you may wish to first contextualize the geodynamics, then the tectonic regional and stratigraphic setting and afterward the local structures. Please consider to change accordingly. As a detailed suggestion, when rewriting the Geological setting, please try to make a first part where you shortly focus on the Messinian-Pleistocene stratigraphy and a second part with the dominant structures. If you wish, you will put together the two in the discussion otherwise the two parts are really the same and the reader cannot distinguish what is the difference between the facts and their interpretation.

Accordingly, to what suggested for the literature review and later in the discussion, figure 1 shall really be implemented. So far I did not see changes. No thrusts, nor offshore geology known is reported on the map. Your aim is not reporting them for the first time but understand their role into the reactivation dynamics, right? With the presentation provided in Fig. 1, instead, you may give the impression that no one has ever worked on this area before and that no tectonics is known onshore. Not even a fault is present on the map on land or onshore. Also a general cross-section of the Marche region shall be presented below the map. I have put some references that can help into the detailed comments in the attached file.

Important. You should shortly describe the geology reported by the Authors on the outcrops related to the Coastal Structure (Conero Promontory?) and Amandola Structure.

DATASET (NEED OF METHODS)

Well done by putting Figure 3 in the paper.

Despite the other changes, which I have appreciated, this section is still a mixture of methods and results with some interpretations within (Fig. 9).

A suggestion on how to make this section better. After you have mentioned the seismo-stratigraphic horizons and how they look like in seismic lines (still missing part), you can make correspond to the regional unconformities cropping out (cf. Artoni). Later on, please describe the general characteristics of the seismic units that you later show in detail, by describing their seismic facies.

Notably, the seismic lines you are later describing, are presented in TWT (s) but they are discussed in meters. So, would not be more conservative reporting the thickness in seconds rather than meters? Otherwise it means you have calibrated that with the wells. In the methods section, that is really needed in the paper, you may wish to write how you compare the thickness of seismic units with that of lithostratigraphic units, which also depends on the velocity model applied.

RESULTS (shift here your descriptions in methods)

More than results, at the present state, this section was presented a summary that would also fit for the conclusions...that's to early in the paper and it is not how results are commonly presented. I am not used to this type of organization of the manuscript, where you put together methods and results with some interpretation in between and then a summary. I would suggest you uniform to classic IMRD organization of the paper with Introduction Methods, Results and Discussion.

Going into the core of the changes, I suggest you to do:

From previous review (now modified): "Once having reorganized the results, if you will show an improved version of Figure 9 (perhaps with isopachs) and a panel including all the transects, the reader will be able to correlate the sectors and the structures, that you have synthetized in this paragraph."

Important. In the results, you should dedicate a figure to some details of the seismic lines that helps constraining the envisaged normal fault characteristics and the growth strata. I have furnished some examples on how to contextualize that also in the discussion.

In the cross-sections I see thrusts, but later in figure 9 those faults are marked as normal faults. This is clearly a restoration model of their distribution. Further there are (E)NE-oriented fault structures that can be observed only on strike seismic lines (available on VIDEPI but not presented in the paper, e.g. Line B-440). Can you show these NE-oriented structures in a separate panel?

To support your interpretation can you provide details of the stratigraphic contacts related to the rotated normal faults and to the eventual growth strata?

In the first review round, I had already suggested that the interpretative seismic lines could possibly be put in one panel to finally provide a clue of the subsurface interpretation in support of the reactivation of pre-existing faults and/or of the interaction between relatively deeply seated and shallower faults accommodating shortening. Worthy of note, I am used to appreciate clean data with line drawings, sketched interpretation just below. Consider when make review to apply this changes.

By the way, the ramp of this structure has a displacement larger in the front than in the rear (ie. where it has a very tiny displacement). This is also true for Fig. 4. You possibly have disharmonic folding going on here. What is the non-coloured part? if they are the carbonate succession please specify it.

DISCUSSION

I suggest to subdivide the interpretation of the regional structures, as they are (results), from the interpretation of their temporal evolution and reactivation through the Pliocene and Quaternary (first paragraph of the discussion).

What you should clarify in the discussion is the geological evidence that allows you to explain the Coastal Structure as a reactivated transpressive feature of the Apennine thrust front at given times. By the way, you should compare your structure with the one traced for example by Artoni 2013 and references therein. Both for the Coastal Structure and for the Amandola structure I have provided in the attached PDF file references.

Further, when presenting figure 9 (that should be in the discussion and not in the results), you show transversal faults which could not easily be presented without showing strike (NW-oriented) seismic lines. I have seen that they are available on VIDEPI (see comment above). Showing them can just help into a clarification of those structures. I still find this figure promising although the reconstruction of the Early Pliocene pre-thrusting normal faults affecting the Marche region is still too weak and needs to be much improved. This figure should go into the discussion anyway. So, for the moment it is out of place. Provided the numerous data available, you could try to make a more detailed map with the thickness distribution of the Early Pliocene deposits?

Important. You are mentioning normal faults but actually in the transects presented they are not occurring as such. Here you make normal faults but you have documented only a normal fault along transect 11 between sector a) and b). As no details are reported about this normal faults, this regional fault map seems like not supported by the reported evidence.

Reactivation topic. I also find very promising the potentiality of applicability of the examples that you have provided on the interaction between inherited extensional structures and the subsequent compressive structures that the paper deals with. Once described in the results the growth strata (in the results), then you can discuss them here as suggested more extensively in the attached PDF file.

Fig. 10 must be improved together with the reinterpretation of seismic lines or it shall explain the structural issues related to the rooting of thrusts which is differently interpreted from one transact to another. If in your interpretation you mean that the pop up structures and their related reverse faults cross-cutting the Messinian deposits are rotated and deformed normal faults, you should then explain that theory. Maybe you can trace better the faults and show their details in the transects.

Fig. 12: is that a repetition from fig. 10? In the last figure there is no graphical support for the seismicity presented in the description as written in the sentences citing the figure and in its caption. There must have been a graphical issue.

A suggestion. At the end of the discussion, you can recall the general and regional implications of your work with a few sentences that contextualize it also in the progress of knowledge of the Apennines. You can set the ground for a take home message that could get into how and what is different in your results or interpretation with respect to previous literature.

CONCLUSIONS

A short reminder as of a take home message and implications would be welcome here.

MINOR COMMENTS

English check required.

I have still found some inconsistencies (e.g., Conero Promontory) and I have carefully reviewed both text and figures, providing many suggestions of change that I expect will be useful to you.

At its current stage, I find this work not yet suitable for the public of Geosciences but after a renewed major review of both text and figures, it should be fine. 

What about Marche Adriatic Structural High  or Marche Adriatic Plateau? Is this a new name of a new geological object?

I require no anonimity and wish that all my comments are forwarded to the Authors. I hope that my review is received as a constructive indication, that may assist the Authors to achieve an even clearer paper, and the Editors in formulating a decision.

Sassari University

Author Response

COMMENTS from REVIEWER 2 with ANSWERS

Dear Authors,

thanks for your second version of manuscript, which deals with the subsurface information from the onshore to offshore southern part of the Marche Apennines. I recognize that by following part of the suggested changes, you have implemented the text, especially in the geological setting. The topic is of great interest to anyone who deals with fault reactivation and foreland basins. You also made some progress concerning the description of the cross-sections and maps. However, in my opinion there are still major changes to the structure of the paper that will help refining methods, results and discussion in order to get to a sound geological model of the area related to the middle Pliocene - earliest Quaternary tectonics. Importantly, the paper has to present a take home message that has to deal with the existing literature on a regional scale. Further, as it deals with reactivation of relatively recent inherited structures, it has to be based on a strong syn-sedimentary evidence.

The suggestions provided in the first review round, where partially skipped. As a result, figures still need to be re-edited not just to follow the journal standards but also to reach a geological description that can support the aimed interpretation regarding the recent structural evolution of the area. I have to remark that their order shall be reviewed following the suggestions in the second reviewed PDF attached.

            R: Dear Reviewer,

Unfortunately seems that you did not check the final revised text of our paper including also the revised English and revised Figures. Some figures have been mixed up between the track-change and the final text file (!?). In the pdf you sent us back there is the new Figure 3 (stratigraphic column), which was not in the track – change file. The Figures of the transects are new, others are the old figure (as in the track-change file) and the text is the one before the English revision. The correct text with correct figures is the “revised text final” that we uploaded in the Journal system.

Furthermore, most of the observations are completely new and they have nothing to do with the observation and advices from the first round of revision. It seems the revision for a new submission. But this IS NOT a new submission, so we cannot take into account all these latter observations.

Nevertheless, we thank you for further accurate revision and accept several suggestions where our revision was not complete or sufficient. Some of the revision are worthless due to the wrong file used.

The order of the mentioned figures has been already changed in the final text file.

With pleasure, I have dedicated care again into commenting again throughout the whole paper. I hope that this effort can help you highlighting the results of your research. You replied that you have accepted that comment but I still see the initial structure which is not the classical Introduction, Geological setting, Methods and materials, Results, Discussion, which helps also reading the paper. Beside the structure of the paper, that is presented with an unusual mixture/transition of methods and results with interpretation maps into the results, I was concerned about the change of the structural interpretation model among the different geological transects presented, for example regarding the geometry of the thrusts.

            R: Beside the titles of the paragraphs (we also revised them), we carefully revised the organization of the text distinguishing between Methods and Results. So there is no possibility of misunderstandings for the Readers.

There is not change of the structural interpretation model among the different geological transects presented. If the reviewer refers to the Amandola structure, it terminates and is not more present in some transects

ABSTRACT

I would focus the beginning of this abstract on the structural problem you aim to show: positive vs negative reactivation of inherited structures in fold and thrust belts or in the Apennines. Here you present an example from the external domain of the Marche Apennines, which display interesting reactivation examples from the subsurface geology explored.

R: The Abstract was improved following “literally” the suggestions. Now it is better focused for general public

INTRODUCTION
The Introduction needs to be a little more improved. In this frame, beside the local interest of the southern Marche structures, you should contextualize your aim into the foreland tectonics of the Apennines and try to make a general outcome out of your results (e.g., on the reactivation or deformation of preexisting structures). Those can be your guidelines in the discussion too.

R: We believe that it was already done

Another point you can better exploit is regarding the presentation of the previous Author work. I have appreciated that you have increased the number of citations. However, they need to be explained and contextualized. Numbering them in loads is not suitable as it does not help making distinction between the different contributions from the different authors. This is relevant as it helps focusing the introduction to the problem that you want to solve and it later helps guiding the discussion towards a more cutting-edge analysis of the regional structures. Have a look at the comments for details.

R: Thanks. Now we made a more accurate distinction between the contributions of previous works, with groups of Papers by topics. Nevertheless, we don’t want to go into details on the single contribution. Citations are needed, but we don't want to make  a Review.

GEOLOGICAL SETTING

The structure of the geological setting is unusual too. As focusing into a geological introduction goes from a larger point of view to a more restricted area of observation, you may wish to first contextualize the geodynamics, then the tectonic regional and stratigraphic setting and afterward the local structures. Please consider to change accordingly. As a detailed suggestion, when rewriting the Geological setting, please try to make a first part where you shortly focus on the Messinian-Pleistocene stratigraphy and a second part with the dominant structures. If you wish, you will put together the two in the discussion otherwise the two parts are really the same and the reader cannot distinguish what is the difference between the facts and their interpretation.

R: OK. We thought that, continuing from Introduction, it was more focused to start with the studies area, but we accept the reviewer’s suggestion and now the general, continental frame is at the top.

About the focus on Messinian-Pleistocene stratigraphy and second part on structures, we believe that the Tectono-stratigraphic evolution of this area has intimate correlations that we don't think is useful or necessary to distinguish in detail.

Accordingly, to what suggested for the literature review and later in the discussion, figure 1 shall really be implemented. So far I did not see changes. No thrusts, nor offshore geology known is reported on the map. Your aim is not reporting them for the first time but understand their role into the reactivation dynamics, right? With the presentation provided in Fig. 1, instead, you may give the impression that no one has ever worked on this area before and that no tectonics is known onshore. Not even a fault is present on the map on land or onshore. Also a general cross-section of the Marche region shall be presented below the map. I have put some references that can help into the detailed comments in the attached file.

R: This comment shows that the reviewer has not seen or considered the new figure 1 included in the final edited text file. We decided to insert only the main first order structure (M.ti Sibillini thrust), because the figure 1, witch includes the geological map, does not allow to  add many structural elements at this scale in our opinion. More detailed structural maps (without geology) are cited in the text included one produced by ourselves (Chicco et al 2019), as underlined by the reviewer. That’s it.

Important. You should shortly describe the geology reported by the Authors on the outcrops related to the Coastal Structure (Conero Promontory?) and Amandola Structure.

R: This is something not included in the first revision. Other cited papers worked on it.

DATASET (NEED OF METHODS)

Well done by putting Figure 3 in the paper.

Despite the other changes, which I have appreciated, this section is still a mixture of methods and results with some interpretations within (Fig. 9).

R: Now we better explained and better split Methods (3. Dataset and working methods) and Results (4. Results by wells and seismic profiles interpretation). This latter chapter is sub-divided in several paragraphs.

A suggestion on how to make this section better. After you have mentioned the seismo-stratigraphic horizons and how they look like in seismic lines (still missing part), you can make correspond to the regional unconformities cropping out (cf. Artoni). Later on, please describe the general characteristics of the seismic units that you later show in detail, by describing their seismic facies.

R: We think this is not relevant for our paper (and also not required in the first revision)

Notably, the seismic lines you are later describing, are presented in TWT (s) but they are discussed in meters. So, would not be more conservative reporting the thickness in seconds rather than meters? Otherwise it means you have calibrated that with the wells. In the methods section, that is really needed in the paper, you may wish to write how you compare the thickness of seismic units with that of lithostratigraphic units, which also depends on the velocity model applied.

R: As suggested in the previous revision we added a fig (Figure 3) and we put a stratigraphic column with these kind of correspondence (sec. vs metres )

RESULTS (shift here your descriptions in methods)

More than results, at the present state, this section was presented a summary that would also fit for the conclusions...that's to early in the paper and it is not how results are commonly presented. I am not used to this type of organization of the manuscript, where you put together methods and results with some interpretation in between and then a summary. I would suggest you uniform to classic IMRD organization of the paper with Introduction Methods, Results and Discussion.

R: Thank you. We better re-organized this part and w erealized that the previous “Resuts” were only a little portion of our Results. Now this is a paragraph within Chap. 4 (Results) only relative to Muddle-Lape Pliocene-Quaternary

Going into the core of the changes, I suggest you to do:

From previous review (now modified): "Once having reorganized the results, if you will show an improved version of Figure 9 (perhaps with isopachs) and a panel including all the transects, the reader will be able to correlate the sectors and the structures, that you have synthetized in this paragraph."

R: We integrated and explained in the new text how we realized Fig. 9 and why we do not realize a map with isopachs.

Important. In the results, you should dedicate a figure to some details of the seismic lines that helps constraining the envisaged normal fault characteristics and the growth strata. I have furnished some examples on how to contextualize that also in the discussion.

R: We did not accept this observation because it is new and above all it would weigh down the description. However, we think we have provided and documented the basic informations.

In the cross-sections I see thrusts, but later in figure 9 those faults are marked as normal faults. This is clearly a restoration model of their distribution.

R: Figure 9 is not only built based on the data of the transects but also using wells. data. In the transects the post-middle Pliocene compressive deformation prevails, while in figure 9 we want to show and highlight the previous deformation (before last compression).

 Further there are (E)NE-oriented fault structures that can be observed only on strike seismic lines (available on VIDEPI but not presented in the paper, e.g. Line B-440). Can you show these NE-oriented structures in a separate panel?

R: We have explained in the text what this fault corresponds to.

To support your interpretation can you provide details of the stratigraphic contacts related to the rotated normal faults and to the eventual growth strata?

R: We think it is explained in the text. Deposits belonging to pre-compressive tectonic are indicated as transgressive and plane-parallel, those subsequent have a pinch-out geometry on growing structures

In the first review round, I had already suggested that the interpretative seismic lines could possibly be put in one panel to finally provide a clue of the subsurface interpretation in support of the reactivation of pre-existing faults and/or of the interaction between relatively deeply seated and shallower faults accommodating shortening. Worthy of note, I am used to appreciate clean data with line drawings, sketched interpretation just below. Consider when make review to apply this changes.

R: This was done within the first revision. In fact we produced two Plates as Supplementary materials: the first is a panel with alla the interpreted transects in the cotìrrect geographic position. The second provide the cleaned seismic profiles. We don’t know why the Assistant Editor did not give you the access to these materials! ma abbiamo preferito non inserire il pannello direttamente nel testo descrittivo  per semplicità di esposizione. We preferred not to insert the panel directly in the descriptive text for simplicity of presentation and to immediately have the figure close to its description. But in the Plates you will see better due to a larger scale!!!

By the way, the ramp of this structure has a displacement larger in the front than in the rear (ie. where it has a very tiny displacement). This is also true for Fig. 4. You possibly have disharmonic folding going on here. What is the non-coloured part? if they are the carbonate succession please specify it.

R: We explained in the text that Messinian deposits are marginally involved in the compressive deformation due to the detachment with respect to the deposits above.

DISCUSSION

I suggest to subdivide the interpretation of the regional structures, as they are (results), from the interpretation of their temporal evolution and reactivation through the Pliocene and Quaternary (first paragraph of the discussion).

What you should clarify in the discussion is the geological evidence that allows you to explain the Coastal Structure as a reactivated transpressive feature of the Apennine thrust front at given times. By the way, you should compare your structure with the one traced for example by Artoni 2013 and references therein. Both for the Coastal Structure and for the Amandola structure I have provided in the attached PDF file references.

R: We have explained in the text similarities and differences compared to what is reported by other authors in our area and outside it.

Further, when presenting figure 9 (that should be in the discussion and not in the results), you show transversal faults which could not easily be presented without showing strike (NW-oriented) seismic lines. I have seen that they are available on VIDEPI (see comment above). Showing them can just help into a clarification of those structures. I still find this figure promising although the reconstruction of the Early Pliocene pre-thrusting normal faults affecting the Marche region is still too weak and needs to be much improved. This figure should go into the discussion anyway. So, for the moment it is out of place. Provided the numerous data available, you could try to make a more detailed map with the thickness distribution of the Early Pliocene deposits?

R: We did not accept this observation because it is new. Nevertheless, we believe our expalnations within Methjods and Results are sufficient.

Important. You are mentioning normal faults but actually in the transects presented they are not occurring as such. Here you make normal faults but you have documented only a normal fault along transect 11 between sector a) and b). As no details are reported about this normal faults, this regional fault map seems like not supported by the reported evidence.

R: In the transepts there are few normal faults because we have said that they have been inverted or decapitated by the subsequent compressive deformation. Nevertheless,  we have indicated where the remnants of these faults are.

Reactivation topic. I also find very promising the potentiality of applicability of the examples that you have provided on the interaction between inherited extensional structures and the subsequent compressive structures that the paper deals with. Once described in the results the growth strata (in the results), then you can discuss them here as suggested more extensively in the attached PDF file.

R: Another new observation! We believe we have sufficiently discussed the topic.

Fig. 10 must be improved together with the reinterpretation of seismic lines or it shall explain the structural issues related to the rooting of thrusts which is differently interpreted from one transact to another. If in your interpretation you mean that the pop up structures and their related reverse faults cross-cutting the Messinian deposits are rotated and deformed normal faults, you should then explain that theory. Maybe you can trace better the faults and show their details in the transects.

R: We can take this into consideration for future developments of our paper, together with other new and extra observations with respect to the first revision.

Fig. 12: is that a repetition from fig. 10? In the last figure there is no graphical support for the seismicity presented in the description as written in the sentences citing the figure and in its caption. There must have been a graphical issue.

R: It is not a mere repetition of Fig. 10 but it is a simplified scheme that introduces the kinematics of the structures and seismotectonic correlations.

A suggestion. At the end of the discussion, you can recall the general and regional implications of your work with a few sentences that contextualize it also in the progress of knowledge of the Apennines. You can set the ground for a take home message that could get into how and what is different in your results or interpretation with respect to previous literature.

R: Thank you for this advice. We now reorganized and better explained our fondongs with respect to previous works.

CONCLUSIONS

A short reminder as of a take home message and implications would be welcome here.

R: Thank you for this advice. We now reorganized and better explained our fondongs with respect to previous works.

MINOR COMMENTS

English check required.

R: English text completely revised by mother tongue. The version of the text used by the reviewer predates this review

I have still found some inconsistencies (e.g., Conero Promontory) and I have carefully reviewed both text and figures, providing many suggestions of change that I expect will be useful to you.

R: Thanks for the suggestions. As far as possible to this state of work, and based on our data, our working methods, our knowledge and our scientific convictions we have done so.

At its current stage, I find this work not yet suitable for the public of Geosciences but after a renewed major review of both text and figures, it should be fine. 

R: We made many new revisions. Unfortunately the Reviwer did not notice some of the revisions we had done before, after the first revision….

What about Marche Adriatic Structural High  or Marche Adriatic Plateau? Is this a new name of a new geological object?

R: This is our name for something partially known, bat well highlighted in its meaning by our fig. 9.

I require no anonimity and wish that all my comments are forwarded to the Authors. I hope that my review is received as a constructive indication, that may assist the Authors to achieve an even clearer paper, and the Editors in formulating a decision.

R: Nobody told us about your identity officially. If you want to be clearly cited with your name as a Reviewer of this paper, please write me. Otherwise we will Thank generally the Reviewers.

Sassari University

Round 3

Reviewer 2 Report

Dear Authors,

by reading your replays and complaints, I have noticed that we have come to misunderstandings. This is sad as I have seen you have not positively judged my detailed constructive comments. If I have asked new things is because the paper in its first round was needing much changes to be done (and were not performed as suggested) that in my point of view, in the second round still there were major issues and I could not avoid to ask you. You know there is no personal issues other than helping each other in making and communicating geology at the best we can.

By talking with one of the editors, I also came to know that I did not receive the latest version of your submitted manuscript file but a more immature version. 

Although I would have performed more of the suggested changes, I recognized that the manuscript has increased in soundness and, if for the editors is ok, this paper is fine.

Even if I have dedicated a lot of time to review your paper twice, I do not particularly require to be in the acknowledgments. 

My best and sincere wishes